

# Modeling considerations for research on Ocean Alkalinity Enhancement (OAE)

Katja Fennel[1], Matthew C. Long[2], Christopher Algar[1], Brendan Carter[3], David Keller[4], Arnaud Laurent[1], Jann Paul Mattern[5], Ruth Musgrave[1], Andreas Oschlies[4], Josiane Ostiguy[1], Jaime B. Palter[6], Daniel B. Whitt[7]

[1]Department of Oceanography, Dalhousie University, Halifax, Nova Scotia, Canada
[2]National Center for Atmospheric Research, University Corporation for Atmospheric Research, Boulder, Colorado, USA
[3]Pacific Marine Environmental Laboratory, National Oceanic and Atmospheric Association, Seattle, Washington, USA
[4]Marine Biogeochemical Modelling, GEOMAR Helmholtz Centre for Ocean Research Kiel, Kiel, Germany
[5]Ocean Sciences Department, University of California Santa Cruz, Santa Cruz, California, USA
[6]Graduate School of Oceanography, University of Rhode Island, Narragansett, Rhode Island, USA
[7]Earth Science Division, NASA Ames Research Center, Moffett Field, California, USA

*Correspondence to*: Katja Fennel (Katja.Fennel@dal.ca)

**Abstract.** The deliberate increase of ocean alkalinity (referred to as Ocean Alkalinity Enhancement or OAE) has been proposed as a method for removing $CO_2$ from the atmosphere. Before OAE can be implemented safely, efficiently, and at scale several research questions have to be addressed including: 1) which alkaline feedstocks are best suited and in what doses can they be added safely, 2) how can net carbon uptake be measured and verified, and 3) what are the potential ecosystem impacts. These research questions cannot be addressed by direct observation alone but will require skillful and fit-for-purpose models. This chapter provides an overview of the most relevant modeling tools, including turbulence-, regional- and global-scale biogeochemical models, and techniques including approaches for model validation, data assimilation, and uncertainty estimation. Typical biogeochemical model assumptions and their limitations are discussed in the context of OAE research, which leads to an identification of further development needs to make models more applicable to OAE research questions. A description of typical steps in model validation is followed by proposed minimum criteria for what constitutes a model that is fit for its intended purpose. After providing an overview of approaches for sound integration of models and observations via data assimilation, the application of Observing System Simulation Experiments (OSSEs) for observing system design is described within the context of OAE research. Criteria for model validation and intercomparison studies are presented. The article concludes with a summary of recommendations and potential pitfalls to be avoided.

## 1 Introduction

Ocean Alkalinity Enhancement (OAE) refers to the deliberate increase of ocean alkalinity, which can be realized either by removing acidic substances from or adding alkaline substances to seawater. OAE is receiving increasing attention as a method for removing $CO_2$ from the atmosphere; such methods are referred to as marine Carbon Dioxide Removal (mCDR) technologies (Renforth and Henderson, 2017). Natural analogues to OAE exist (Chapter 4.2). An increase in the alkalinity of





seawater leads to a repartitioning of its dissolved carbonate species with a shift toward bicarbonate and carbonate ions (Zeebe

and Wolf-Gladrow 2001, Renforth and Henderson 2017), leading to a reduction in the aqueous $CO_2$ concentration and thus the partial pressure of $CO_2$ ($p$CO$_2$; Chapter 2). Since exchange of $CO_2$ between the ocean and atmosphere occurs when the surface ocean $p$CO$_2$ is out of equilibrium with that of the atmosphere, a lowering of the ocean's $p$CO$_2$ will lead to a net ingassing of atmospheric $CO_2$ (i.e., an increase in $CO_2$ uptake by the ocean or a decrease in outgassing due to OAE). This would increase the oceanic and decrease the atmospheric inventories of inorganic carbon, in other words, it would result in

mCDR. In contrast to other mCDR technologies, OAE does not exacerbate ocean acidification (Ilyina et al. 2013). In fact, an increase in ocean alkalinity counteracts acidification, and while subsequent net uptake of atmospheric $CO_2$ largely restores pH to its pre-perturbation value, there is potential for OAE deployment to mitigate acidification impacts near injection sites (Mongin et al. 2021).

Several important research questions should be addressed before implementing OAE as an mCDR technology at scale. These

include: 1) which alkaline substances are best suited and in what doses can they be added reliably while avoiding precipitation of calcium carbonate (which would decrease alkalinity and could result in runaway precipitation events), 2) how can changes in alkalinity and net carbon uptake be measured, verified, and reported (referred to as MRV; see Chapter 6) to enable meaningful carbon crediting, and 3) what are the potential ecosystem impacts and how can harm to ecosystems be avoided or minimized while maximizing potential benefits. These research questions cannot be addressed by direct observation alone, but

will require an integration of observations and numerical ocean models across a range of scales. Skillful and fit-for-purpose models will prove valuable for addressing many OAE research questions including the MRV challenge, assessment of environmental impacts, and interpretation of natural analogs.

Ocean models are useful for a broad range of purposes, from idealized models for basic hypothesis testing of fundamental principles to realistic models for more applied uses (see primer on ocean biogeochemical models by Fennel et al. 2022). In the

context of OAE research, this full range of models is applicable. For example, idealized models of particle-fluid interaction can inform us about dissolution and precipitation kinetics at the scale of particles, realistic local-scale models can inform us about nearfield processes in the turbulent environment around injection sites, and larger-scale regional or global ocean models can be used to support observational design for field experiments, to demonstrate possible verification frameworks, and to address questions about global-scale feedbacks on ocean biogeochemistry. A common objective of all of these modeling

approaches is to realistically simulate the spatio-temporal evolution of the seawater carbon chemistry, including alkalinity and dissolved $CO_2$, and attribute that evolution to physical, chemical, and biological processes. Models that are suitable for this purpose will provide spatial and temporal context for properties that can be observed (but at much sparser temporal and spatial coverage than a model can provide) as well as estimates of properties and fluxes that cannot be directly observed but may be inferred because of known mechanistic relationships or patterns of correlation. Applications of realistic models rely on them

being skillful and accurate, requiring that they include parameterizations of the relevant processes, and that they are constrained



by observations that contain sufficient meaningful information. An array of methods that combine observations and models in optimal ways are available. Application of such methods is referred to as data assimilation and provides the most accurate estimates of biogeochemical properties and fluxes (see Fennel et al. 2022 for fundamentals and code examples).

Model applications for OAE research include the following four general types:

- Hindcasts are model applications where a defined time period in the past was simulated. They can be unconstrained—in the sense that no observations are fed into the model except for initial, boundary, and forcing conditions—or constrained, where observations inform the model state via data assimilation. The latter are also referred to as optimal hindcasts or reanalyses.

-  Nowcasts/forecasts are similar to constrained hindcasts but with the simulations carried out up to the present (referred to as nowcasts) or into the future (referred to as forecasts). The latter require assumptions about future forcing and boundary conditions, e.g., from other forecasts, climatology, or assuming persistence.

- Scenarios, or counterfactual simulations, are unconstrained hindcasts or forecasts where one or more aspects of the model is systematically perturbed to assess the effect of the perturbation, for example, in paired simulations with and without OAE one will be a scenario. These can be used to explore even very unlikely situations, which is often required in comprehensive uncertainty and risk assessment.

- Observing System Simulation Experiments (OSSEs) for observing system design use unconstrained and/or constrained hindcasts to evaluate the benefits of different sampling designs and optimize deployment of observational assets for a defined objective, including tradeoffs between different types of observation platforms.

Successful implementation of models to support OAE research and MRV is challenging because of the general sparseness of relevant biogeochemical observations, especially when considering the small spatial and temporal scales of OAE field experiments, and the limited lab, mesocosm, and field trial data available to date for model parameterization. Further, models are built at a process level and integrated to reveal behavior at the emergent scale. As such, models comprise a collective hypothesis of the ocean's physical, biogeochemical, and ecosystem function—but it is important to recognize that model formulations of key processes related to OAE remain uncertain. It may well turn out that parameterizations of the carbonate system, of plankton diversity and trophic interactions, small scale turbulence, submesoscale subduction and restratification processes, and air-sea gas exchange in the current generation of models require improvement to robustly treat OAE-related questions.

The intended scope of this chapter is to provide an overview of the most relevant modeling tools for OAE research with high-level background information, illustrative examples, and references to more in-depth methodological descriptions and further examples. We aim to provide simple criteria and guidance for researchers on the current state-of-the-art of biogeochemical modeling relevant to OAE research, keeping in mind short-term research goals in support of pilot deployments of OAE and



long-term goals such as credible MRV in an ocean affected by large-scale deployment of OAE and possibly other mCDR technologies.

## 2 Modeling approaches

This section provides a brief review of modeling tools available for OAE research with references to more in-depth methodological descriptions and examples, as well as a discussion of which approaches are most applicable to simulating essential processes in different circumstances. The presentation is structured using two complementary organizing principles, the spatial and temporal scales of the problem in Section 2.1 and the biogeochemical and ecological complexity represented by different modeling approaches in Section 2.2. and concludes with a summary of suggested future model development efforts

in Section 2.3.

### 2.1 Modeling approaches across scales

In the nearfield, close to the site of an alkalinity increase, an accurate characterization of the spatio-temporal evolution of alkalized waters requires direct representation or parameterization of fluid and particle physics and seawater carbonate chemistry at scales ranging from micrometers to hundreds of meters, spanning turbulent to submesoscale processes (Section

2.1.1).  In the farfield, covering scales from 10s of meters to 100s of kilometers, where the effect of an alkalinity increase depends less on the details of how the alkalinity was added, or acidity removed, and is instead dominated by ambient environmental processes, local to regional scale models are useful for simulating the impact of alkalinity increases, for verifying the intended perturbations in air-sea exchange of $CO_2$ and in carbonate system variables, and potentially for simulating ecosystem impacts  (Section 2.1.2). Lastly, investigation of the effects of the global ocean's overturning circulation,

impacts on atmospheric $CO_2$ levels, and of Earth system feedbacks resulting from deployment of OAE and other mCDR technology at scale requires global modeling approaches (Section 2.1.3).

### 2.1.1 Particle scale to nearfield/turbulence scale (μm to km scales)

Small-scale modeling approaches cover the range from μm-size particles to the turbulent- and submeso-scales in the nearfield of alkalinity additions. Simulating processes on these scales allows one to address questions about how turbulent mixing dilutes

and disperses alkalized water and how it affects the settling, aggregation, disaggregation, precipitation, and dissolution of suspended particles. Nearfield modeling has an important role to play in guiding the design of deployment strategies that mitigate environmental impacts and meet future permitting requirements, and to support monitoring. During the initial dispersion and dilution phase of an alkalinity increase in the nearfield, the direct impacts on carbonate system variables are greatest, with waters exhibiting the largest elevations in pH and the highest potential for the formation of secondary

precipitates. For particulate alkalinity feedstocks, turbulence close to the deployment site affects dissolution and settling rates,





increasing dissolution and either accelerating or diminishing the settling of sedimentary particles compared to the Stokes settling speed (Fornari et al. 2016).

Distinct approaches to modeling at these scales involve different levels of parametrization and computational expense, with the relative utility of each approach being dependent on the scientific questions at hand. At the smallest scales, Direct
Numerical Simulations (DNS) are the most computationally expensive and specialized class of fluid modeling, as they resolve flows down to the scales at which flow variances dissipate—typically centimeters or smaller in the ocean. Consequently, computational constraints imply that they cannot be run over domains larger than a few meters. DNS are thus integrated over idealized physical domains (i.e., they lack realistic bathymetry) and are suited to investigating fundamental physical processes. For example, multiphase DNS simulations have been used to model the interaction of turbulence with gas bubbles (Farsoiya
et al. 2023) and particles (Fornari et al. 2016). Results from such studies provide an important testbed that can be used to develop parameterizations required in lower resolution models.

A well-established approach to modeling the fluid flow at scales up to about 10 km uses Large Eddy Simulations (LES), a class of model that directly solves the unsteady Navier-Stokes equations down to the largest turbulent scales on a high-resolution grid. Such models parameterize turbulence using a subgrid-scale model (e.g., Smagorinsky 1963). An advantage of
these models is their ability to simulate both an alkalized plume and the environmental turbulence into which the plume emerges. Once alkalized waters enter the surface boundary layer, LES models have an established history of simulating turbulence and mixing that is directly relevant to OAE research (e.g., Mensa et al. 2015, Taylor et al. 2020). An example of an LES simulation of near-surface turbulence dispersing surface deployed alkalinity downwards is illustrated in Figure 1, where a physical model (Ramadhan et al. 2020) has been coupled to a carbonate solver (Lewis et al. 1998). To date, LES models
have rarely been coupled to biogeochemical models due to the computational expenses involved, though their inclusion may be increasingly feasible (Smith et al. 2018, Whitt et al. 2019). As LES simulate flow physics at scales ranging from 10-10,000 m, they do not explicitly resolve the microscales of fluid motion and chemical reactions at particle scales. Nevertheless, the parameterizations of such processes can be included; for example, Liang et al. (2011) used models of bubble concentration and dissolved gas concentration in an LES to examine the influence of bubbles on air-sea gas exchange.



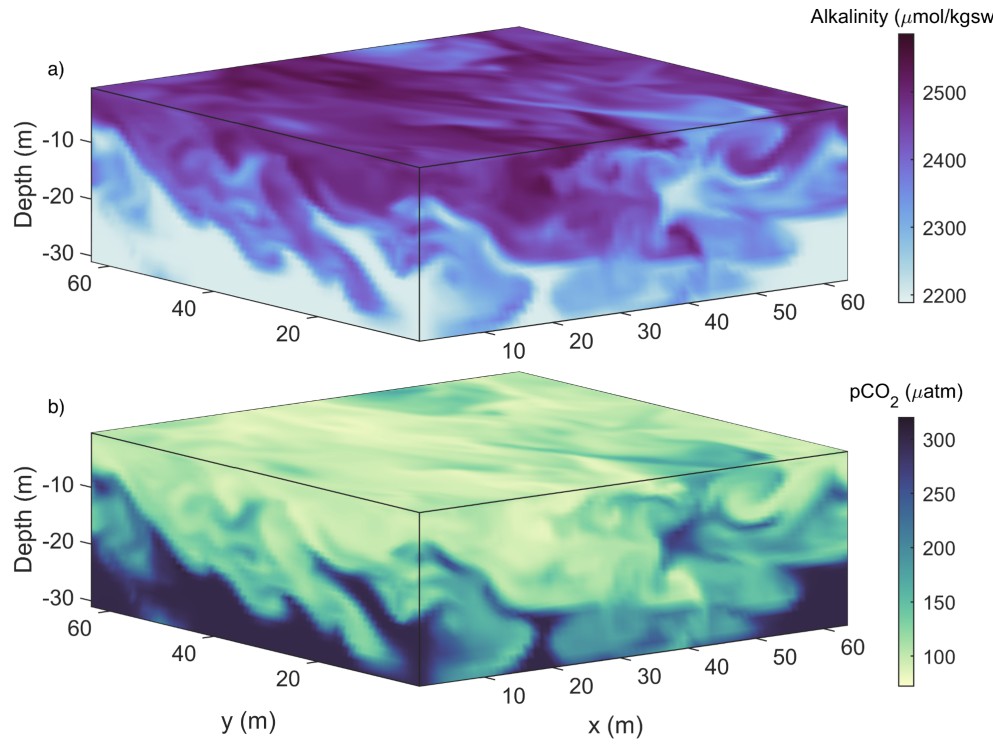

**Figure 1:** LES of near surface turbulence coupled to a carbonate system solver. Alkalinity is added at a rate of 4 μmol kgsw$^1$ m$^{-2}$ s$^{-1}$ for 20 minutes to the top grid cell at the start of the simulation. Turbulence, generated by surface wind stress and cooling, sets the rate at which it mixes downwards (a) along with associated waters of lowered $p$CO$_2$ (b). Turbulent plumes and eddies lead to inhomogeneities in water properties at scales of tens of meters.

For alkalized plumes associated with outfalls from, for example, wastewater treatment plants, integral models (that assume plume properties such that the governing equations are simplified) have been developed to examine the initial dilution close to jets and buoyant plumes up to kilometer scales (Jirka et al. 1996). These models are highly configurable, enabling specific diffuser configurations as well as the potential to incorporate sediment laden plumes with particle settling (Bleninger & Jirka 2004). Results are commonly accepted for engineering purposes, defining mixing zones, and providing a fast "first look" at diffusion and mixing near an outfall site. However, these models rely on assumptions about the underlying physics of fluid flow (e.g., axisymmetric plumes and simplified entrainment rates) that may not be accurate under general oceanic conditions, and results will not include all effects of irregular bathymetry, finite domain size or arbitrarily non-uniform ambient conditions. Nevertheless, their simplicity makes them very useful. For example, by combining several simple process models for plume dilution, particle dissolution, and carbon chemistry, Caserini et al. (2021) have simulated the initial dilution of slaked lime Ca(OH)$_2$ particles and alkalinity in a plume behind a moving vessel.





Other methods for modeling at this scale include Reynolds Averaged Navier Stokes (RANS) and Unsteady RANS (URANS), wherein fluctuations against a slowly varying or time mean background are parametrized, often using constant (large) eddy diffusivities and viscosities. These approaches are often inaccurate at these scales, resulting in simulations that are too diffusive or lacking processes that are of leading order importance to mixing (Golshan et al. 2017, Chang & Scotti 2004).

There are multiple, potentially interacting sources of uncertainty to consider when evaluating the uncertainty of the application described above. Perhaps best understood but still problematic is the uncertainty that arises from the computational intractability of simulating all the relevant scales in the μm to km range at once, necessitating the different modeling approaches for different scales, with parameterizations to account for unresolved scales and scale interactions. The dissolved carbonate chemistry of seawater is relatively well parameterized (Zeebe and Wolf-Gladrow 2001), but some modest uncertainties arise

from approximations required for computational tractability (Smith et al. 2018). The least understood but potentially dominant source of uncertainty pertains to the representation of the microscale biological, chemical, and physical dynamics of particles, which is an active area of experimental and observational investigation (Subhas et al. 2022, Fuhr et al. 2022, Hartmann et al. 2023). These questions about particles apply to those released in OAE deployments, as well as particles that precipitate from seawater in part due to OAE deployments, and finally the role of ambient biotic and abiotic particles where OAE is deployed.

**2.1.2 Local to regional scales (m to km)**

Local to regional scale models that range in horizontal resolution from tens of meters to hundreds of kilometers are useful for simulating the impact of alkalinity injections beyond the immediate local area, where conditions do not depend on the details of how the alkalinity was added and instead are determined by regional-scale currents and other process, including the potential for biogenic feedbacks. These models are particularly useful to support OAE field experiments, including planning and

observational design, and analysis, integration and synthesis of observations, and to facilitate interpretation of observations from natural analogs. Furthermore, local and regional scale models will likely prove to be indispensable for quantification of OAE effects in research settings, for guiding assessments of its environmental impacts, and for MRV during the potential implementation of OAE. A skillful model can simulate when and where changes in carbonate chemistry and the ensuing anomalies in air-sea $CO_2$ exchange occur and provide an estimate of the spatio-temporal extent of the biogeochemical

properties affected by OAE.

Regional models have distinct advantages over global models in their ability to resolve the spatial scales on which OAE would be applied both experimentally and operationally, and their documented skill in representing coastal and continental shelf processes more accurately (Mongin et al. 2016, Laurent et al. 2021). Examples of regional model applications in the context of OAE include the recent studies by Mongin et al. (2021) and Wang et al. (2023). Mongin et al. (2021) used a coupled

physical-biogeochemical-sediment model tailored to Australia's Great Barrier Reef to investigate to what extent realistic OAE



applied along a shipping line could alleviate anthropogenic ocean acidification on the reef. Wang et al. (2023) used a coupled ice-circulation-biogeochemical model of the Bering Sea to study the efficiency of OAE in coastal Alaska.

Implementation of a regional model in a target domain requires generation of a grid with associated bathymetry, specification of boundary conditions (including atmospheric forcing, information about ocean dynamics along the lateral boundaries of the domain, any fluxes of biogeochemical properties across the air-sea, sediment-water, and land-ocean boundaries, river inputs), and generation of initial conditions within the domain (Fennel et al. 2022). Different circulation models are available for implementation in domains targeted for OAE studies (see, e.g., Table 1 in Fennel et al. 2022), all with distinct strengths and established user communities. Particularly relevant in the context of studying coastal applications of OAE is a model's ability to accurately represent coastal topography, making unstructured grid models and models with terrain-following coordinates particularly attractive. Another feature to be considered is a model's ability to run in two-way nested configurations. In the more widely applied one-way nesting of domains, simulated conditions from a larger scale model (referred to as the parent model) are used to generate the dynamic lateral boundary conditions of a smaller scale, higher resolution model (the child model), which runs off line from the parent model. With two-way nesting, both models run simultaneously and information is exchanged continually along their intersecting boundaries. This allows information generated within the high-resolution child domain (e.g., the spreading distribution of a tracer or alkalinity addition) to be received and propagated by the larger-scale parent model.

One example of a high-resolution local scale model with two-way nested domains is a framework developed for Bedford Basin in Halifax, Canada (Figure 2, Laurent et al., in prep.). The model framework consists of three nested ROMS models (ROMS is the Regional Ocean Modelling System; https://myroms.org, Haidvogel et al. 2008, Shchepetkin and McWilliams 2005). The outermost ROMS domain has a resolution of 900 m and is nested one-way within the data-assimilative global GLORYS reanalysis of physical and biogeochemical properties (Lellouche et al. 2021). Nested within are two models with increasingly higher resolutions of 200 m and 60 m. Depending on the scientific objective to be addressed, the models can be run in one-way and two-way nested mode, where two-way nesting is computationally more demanding. Implementation of dye-tracers within the model (Wang et al. in prep.) allows one to determine dynamic distribution patterns and residence times.



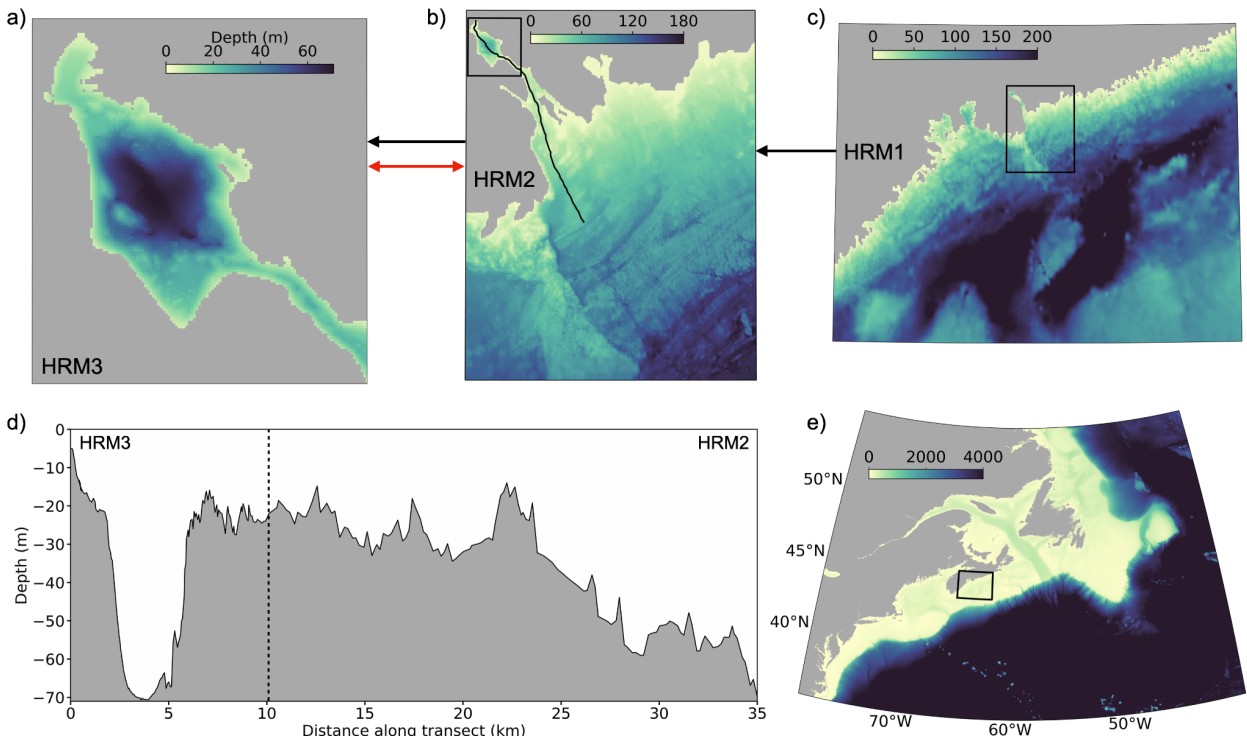

**Figure 2:** Nested configuration of three ROMS models for the Bedford Basin and the adjacent harbor in Halifax Regional Municipality (HRM). a) The highest resolution model (HRM3; 60 m) includes the 7 km-long and 3 km-wide Bedford Basin and The Narrows, a 20-m shallow narrow channel that connects the basin to the outer harbor. b) The larger scale model (HRM2, 200 m) includes Bedford Basin and Halifax Harbor as well as the adjacent shelf. c) The largest-scale model (HRM3, 900 m) covers the central part of the Scotian Shelf as indicated in e). d) bathymetry along a section through HRM3 and HRM2, indicated by the black line in b). Lateral boundaries of HRM3, HRM2, and HRM1 are shown by black boxes in b), c) and e), respectively. Black arrows indicate the information flow between models in one way nesting mode. The red arrow indicates that HRM1 and HRM2 can be run simultaneously with bi-directional flow of information (two-way coupled mode).

### 2.1.3 The global scale

A strength of global ocean models is their capacity to comprehensively represent the global overturning circulation and ocean ventilation. These processes control the time scales over which waters are sequestered in the ocean interior and determine how long surface waters are exposed to the atmosphere and can exchange properties, including $CO_2$, before being injected back into the ocean interior (Naveira Garabato et al. 2017). Similarly, the large-scale overturning circulation and the patterns associated with ventilation are important to consider in the context of deploying OAE at scale, as these patterns exert strong control on the efficiency of OAE at sequestering $CO_2$ (e.g., Burt et al. 2021).



When global ocean models are dynamically coupled with models of the land biosphere and the atmosphere, they are referred to as Earth System Models (ESMs) and can be employed to explore Earth system feedbacks to mCDR. In the case of OAE, the main feedback is the change in atmospheric $p$CO$_2$ and air-sea gas exchange that will result when CDR approaches are implemented at scale. While regional models have to be forced by atmospheric CO$_2$ concentrations, ESMs represent the

atmospheric reservoir and are forced by CO$_2$ emissions into the atmosphere, which then interacts with land and ocean carbon reservoirs. Only the latter approach can account for OAE-induced reductions in the atmospheric CO$_2$ inventory which, in turn, would lead to a systematic reduction in air-sea CO$_2$ fluxes. Regional models and global ocean models that do not explicitly represent the atmospheric CO$_2$ reservoir and instead are forced by prescribed atmospheric $p$CO$_2$ cannot simulate the decline in atmospheric $p$CO$_2$ due to OAE. Depending on the alkaline material applied, there may also be feedbacks associated with

changes in temperature, albedo, nutrient cycles, and biological responses which can be studied with the help of ESMs.

Another important strength of global models relates to the fact that anomalies in air-sea CO$_2$ flux generated by significant OAE deployments will manifest over large spatio-temporal scales because CO$_2$ equilibrates with the atmosphere via gas exchange slowly. Unless equilibrated before the addition or added to seawater that is oversaturated in CO$_2$, alkalinity enhanced waters can be transported far away from injection sites before equilibration is complete (He and Tyka 2023). Consequently, OAE

signals may exit the finite domain of regional models prior to full equilibration with the atmosphere (e.g., Wang et al. 2023). Because global models represent the entire ocean and can be integrated for centuries and longer, they enable full-scale assessments.

A primary challenge for global models, however, is that their horizontal resolution is necessarily limited by computational constraints (see example in Figure 3). Most of the global ocean models contributing the Coupled Model Intercomparison

Project version 6 (CMIP6), for example, have horizontal resolutions of about 1° or roughly 100 km (Heuzé 2021) and do not accurately represent biogeochemical processes along ocean margins (Laurent et al. 2021). Model grid-spacing imposes a limit on the dynamical scales that can be explicitly resolved in the models; this is particularly problematic for coarse resolution global models because mesoscale eddies—i.e., motions on scales of about 10–100 km—dominate the variability in ocean flows (Stammer 1997). Since coarse resolution models cannot resolve mesoscale eddies explicitly, the rectified effects of these

phenomena, including their role in transporting buoyancy and biogeochemical tracers, must be approximated with parameterizations (e.g., Gent and McWilliams 1990).

Notably, the fidelity of the simulated flow in global models, including the imperfect nature of these parameterizations, projects strongly on the model's capacity to accurately simulate ventilation and the associated uptake of transient tracers, such as anthropogenic CO$_2$ or chlorofluorocarbons (CFCs), from the atmosphere (e.g., Long et al. 2021). Biases in the uptake of

transient tracers will also have implications for a model's capacity to faithfully represent the impact of OAE, where the path of alkalinity-enhanced waters parcels in the surface ocean, and their subsequent transport to depth is a key control on the efficiency of carbon removal. Biases in the simulated flow are also an important determinant of the simulated distribution of

biogeochemical tracers in the model's mean state. Hinrichs et al. (2023), for example, demonstrate that inaccuracies in the physical redistribution of alkalinity by the flow is a dominant mechanism contributing to biases in the alkalinity distributions simulated by CMIP6 models.

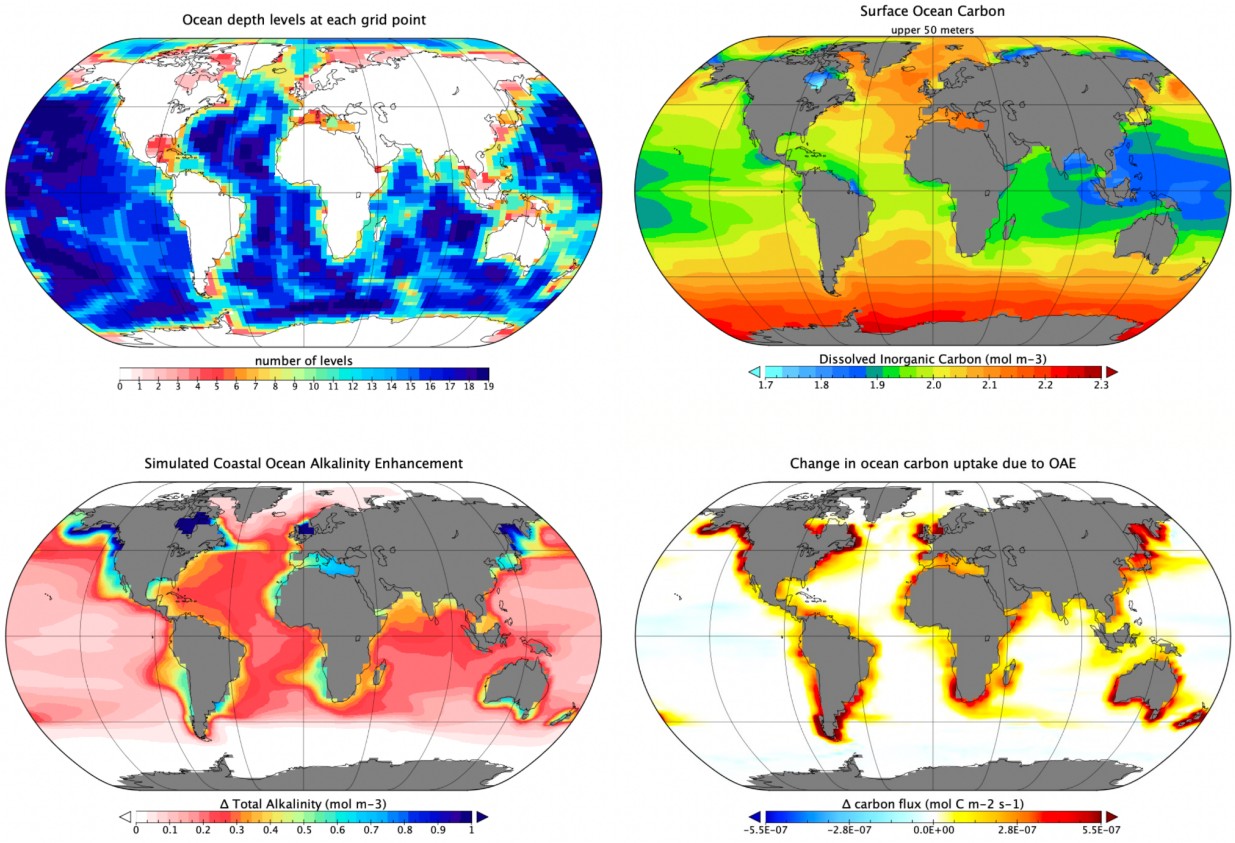

**Figure 3:** Example of Earth System Model properties and output from the University of Victoria Earth System Climate Model (Keller et al., 2012, Mengis et al., 2021) including a) the model bathymetry (depth levels), and b) the simulated present-day dissolved inorganic carbon concentration (mol m$^{-3}$) averaged over the upper 50 m of the ocean. Panels c) and d) show results from a coastal OAE study by Feng et al. (2017) where the change in upper ocean alkalinity (upper 50 m) and the air-sea flux of $CO_2$ are shown relative to the RCP8.5 control simulation. Shown is the Oliv100_Omega3.4 simulation from Feng et al. (2017), where 100 μm olivine grains were added to ice-free coastal grid cells in proportion to RCP 8.5 $CO_2$ emissions (i.e., 1 mol of alkalinity per mole of emitted $CO_2$) until a sea surface aragonite Ω threshold of 3.4 was reached.

Finally, another important challenge associated with global ocean models is the requirement to represent the entire global ocean ecosystem with a single set of model parameters (e.g., Long et al. 2021, Sauerland et al. 2020). In particular, the biological pump is an important control on the distribution of biogeochemical tracers, including alkalinity and DIC. The





magnitude of organic carbon export, and the magnitude of biogenic calcium carbonate export, are important controls on the distribution of alkalinity and DIC at the ocean surface and in the interior (e.g., Fry et al., 2015). These quantities are a product of ecosystem function and, since the global ocean is characterized by diverse biogeography (e.g., Barton et al., 2013), capturing

global variations in the biological pump presents a challenge.

### 2.1.4 Integration across scales

Choosing the appropriate modeling tool for a given OAE-related question requires clarity about the scale of the problem to be addressed and the objectives of the model application. Approaches for OAE vary significantly with respect to the spatial footprint of alkalinity increase. Proposed methods for spreading alkalinity feedstocks at the surface ocean include the addition

of reactive minerals (e.g., CaO, Ca(OH)$_2$ or Mg(OH)$_2$) in ship-propeller washes (e.g., Köhler et al., 2013, Renforth et al., 2017, Caserini et al., 2021) or using other means (e.g., Gentile et al., 2022) along tracks from commercial or dedicated OAE vessels; the addition of less-reactive minerals to corrosive or high-weathering environments (e.g., olivine spreading on beaches or mineral addition to riverine discharge, e.g., Montserrat et al., 2017, Foteinis et al., 2023, Mu et al., 2023); and electrochemically generated point-sources of alkalinity that are discharged as highly alkaline seawater (e.g., House et al., 2009) from existing

facilities (e.g., desalination and wastewater-treatment plants), dedicated facilities (e.g., Wang et al., 2023), or from an array of smaller infrastructure (e.g., grids of off-shore wind turbines). Models for OAE research should represent these footprints of alkalinity increases appropriately for the questions being addressed.

There are research questions that fall relatively neatly into one of the three scale ranges described above in sections 2.1.1 to 2.1.3. For example, consideration of the nearfield effects of different alkalinity feedstocks (e.g., dissolved versus particles) or

analysis of the potential impacts from secondary CaCO$_3$ precipitation due to elevated alkalinity from a point source require models that resolve the scales of turbulent motion. Examination of the change in air-sea CO$_2$ flux due to a broad and diffuse alkalinity increase is less demanding on model resolution and regional scale models are appropriate for this question. Investigation of Earth system feedbacks requires ESMs. However, there also are many aspects of OAE that require a bridging of scales. For example, when considering different deployment methods like discharge from vessels into the ocean surface

boundary layer versus additions made through outfalls via surface or subsurface plumes, modeling requirements vary. In both cases, the resulting biogeochemical response may be affected by dynamics operating in the nearfield, where conditions are sensitive to the deployment method and turbulence has to be considered, and the far-field, where conditions do not depend on the details of how the alkalinity was added and the air-sea flux of CO$_2$ is instead determined by ambient environmental processes. Some interplay among the modeling tools described in sections 2.1.1 and 2.1.2 is likely going to be required. One

straightforward approach would be to parameterize small-scale processes in the larger-scale models.



Another example is the challenge that anomalies in air-sea $CO_2$ flux generated by significant OAE deployments will manifest over large spatio-temporal scales because $CO_2$ equilibrates with the atmosphere via gas exchange slowly. This precludes a comprehensive assessment of perturbation impacts with regional models alone and will likely require the dynamic coupling of high-resolution regional and coarse-resolution global models.

## 2.2 The range of biogeochemical realism & complexity

Application of biogeochemical ocean models for the purposes of OAE research and verification requires reevaluation, and likely further development, of several model assumptions and features related to biogeochemical realism and complexity. For example, the internal sources and sinks of alkalinity are typically not explicitly represented in ocean models; this may become necessary in some circumstances but will be challenging (Section 2.2.1). OAE-related perturbations of alkalinity, other carbonate system properties, and addition of macro- and micronutrients contained in some alkalinity feedstocks may result in biological and ecosystem responses that current biogeochemical models are not capable of representing but that would be relevant for the assessment of environmental impacts of OAE and the verification its CDR efficiency (Section 2.2.2). Furthermore, depending on the environmental setting, sediments can be sources or sinks of alkalinity; these sediment-water fluxes need to be appropriately considered, including the potential impacts of OAE on their magnitude, in order to obtain complete and trustworthy carbon budgets (Section 2.2.3). Other boundary fluxes that require accurate specification are alkalinity inputs from rivers and groundwater (Section 2.2.4) and the air-sea flux of $CO_2$ across the air-sea interface (Section 2.2.5).

### 2.2.1 Representing alkalinity in seawater

Alkalinity is an emergent property that depends on the concentrations of numerous chemical species with distinct internal source and sinks (Chapter 2; Wolf-Gladrow et al., 2007; Middelburg et al., 2020). Skillful simulation of alkalinity in seawater may require explicit representation of its multiple biotic and abiotic sources and sinks, some of which are difficult to constrain. A major process by which alkalinity is consumed is the production of calcium carbonate. In the water column, this is predominantly a biotic process, performed by calcifiers, although "whiting" events, where calcium carbonate precipitates spontaneously from in ambient seawater can be locally important (e.g., Long et al., 2017).

Models vary in the degree of mechanistic sophistication with which biogenic calcification is represented. For example, some models explicitly resolve calcifiers, such as pelagic coccolithophores (e.g., Krumhardt et al., 2017) and foraminifera (Grigoratou et al. 2022) and, in some cases, also benthic corals, foraminifera, or calcifying higher trophic levels and thus can mechanistically account for the associated alkalinity consumption. Alternatively, models can parameterize biotic production of carbonate, and its subsequent sinking and dissolution, as a fraction of organic matter production combined with an assumed remineralization profile (e.g., Schmittner et al. 2008; Long et al., 2021). Dissolution of carbonate minerals produces alkalinity, at the sediment surface and in the water column as carbonate particles sink. This can be represented with first-order abiotic



dissolution kinetics with a dependence on the saturation state of ambient water in the water column (e.g., Sulpis et al., 2021), in the sediments (e.g., Emerson & Archer, 1990) or in micro-environments in aggregates or organisms (Barrett et al., 2014)

with systematic differences for different crystal structures, aragonite and calcite (Morse et al., 1980).

Production of alkalinity occurs via uptake of nitrate or nitrite by photoautotrophs, while remineralization consumes alkalinity when happening aerobically but generates alkalinity when occurring anaerobically, e.g. via denitrification (Fennel et al. 2008). Biotic production and consumption of alkalinity is stoichiometrically coupled to the release or uptake of nutrients and carbon, where non-Redfield processes such as nitrogen fixation or denitrification need to be specifically considered in the

stoichiometric relationships (Paulmier et al., 2009).

Spontaneous precipitation of carbonate minerals in pelagic environments could occur when seawater is highly oversaturated with respect to carbonate (Moras et al. 2022) but is, to the best of our knowledge, not yet included in ocean models. When simulating OAE approaches that may generate high oversaturation with respect to carbonate, spontaneous precipitation of carbonates needs to be considered, especially when condensation nuclei are present. Appropriate approaches will have to be

developed, e.g., using near-field models to mechanistically represent this process and a meta-model approach to develop parameterizations that are suitable for far-field and larger-scale models.

Organic compounds produced within the ocean or originating from land can also act as proton acceptors and contribute organic alkalinity (e.g., Koeve and Oschlies 2012, Ko et al. 2016, Middelburg et al. 2020) and will impact the carbonate system, the partial pressure of $CO_2$ and thus the air-sea $CO_2$ flux. Commonly, the contribution of organic alkalinity is deemed small enough

in oceanic environments to be negligible, but this assumption should be reconsidered in the context of OAE, especially for coastal CDR deployments where the organic contribution to alkalinity is thought to be larger. To the best of our knowledge, models do not account for organic alkalinity. A better quantitative understanding of organic contributions to alkalinity is likely needed to parameterize or mechanistically represent its contribution in models. Similarly, it may be important in the context of mineral OAE deployments to account for local variations in $[Ca^{2+}]$ and $[Mg^{2+}]$ to accurately estimate the $p$CO$_2$ anomalies

generated by different OAE feedstocks. While these constituents have very long residence times in the ocean, and are hence commonly assumed to vary conservatively in proportion to salinity, variations in their relative abundance has an impact on the thermodynamic equilibrium coefficients used to solve seawater carbonate chemistry (Hain et al., 2015).

### 2.2.2 Representing biological and ecological processes

A key question related to OAE is whether changes in carbonate chemistry induce differential responses in organisms. In the

pelagic zone, OAE might shift the phytoplankton community composition, for example, due to distinct physiological sensitivities of different groups (e.g., Ferderer et al. 2022). Further, if OAE is accomplished via rock dissolution, carbonate versus silicate rock may impact the relative balance between phytoplankton functional groups (PFTs) such as calcifiers and diatoms, and changes in Mg and Ca ratios may also influence calcification (Bach et al., 2019). Additionally, ancillary





constituents specific to particular feedstocks may have biological activity. Silicate rocks include bioreactive metals such as Fe,
a micronutrient with the capacity to stimulate phytoplankton growth, and others that are can be toxic when occurring in high
concentrations, such as Ni and Cu, and may adversely impact phytoplankton and reduce primary productivity (Bach et al.,
2019). The bioreactivity of these metals may be difficult to simulate in models as their dissolved concentrations can be partially
mediated by complexation with organic ligands (Guo et al., 2022). Physical impacts of OAE feedstocks may also have
important biological impacts through changes in the propagation of light in the surface ocean, and direct exposure to mineral
particles may have additional impacts, e.g., on zooplankton through particle ingestion (Harvey, 2008; Fakhraee et al., 2023).
Effects of OAE on plankton have the potential to propagate to higher trophic levels through marine food webs as the magnitude
and quality of net primary productivity shifts and trophic energy transfer is altered accordingly.

Simulating this full collection of processes in models is challenging. Dominant modeling paradigms for simulating planktonic
ecosystems include PFT- and trait-based models (e.g., Negrete-Garcia et al., 2022). In these systems, physiological sensitivities
are parameterized according to transfer functions that modulate rate processes—growth, for instance—on the basis of ambient
environmental conditions. Nutrient limitation of growth is often represented using Michaelis–Menten kinetics wherein growth
rates decline as nutrients concentrations become limiting. State-of-the-art ESMs represent PFTs with multiple nutrient co-
limitation, which is essential to effectively simulate plankton biogeography of the global ocean. Diatoms, for example, are
capable of high growth rates, enabling them to outcompete other phytoplankton under high-nutrient conditions, but their range
is restricted to high latitudes and upwelling regions where there is sufficient silicate. If OAE were to modulate the concentration
of constituents represented by multiple nutrient co-limitation models, it is possible such models could simulate the
phytoplankton community response—though it's important to consider whether the models provide representations that are
sufficiently robust for the magnitude of OAE-related perturbations. In some cases, models are missing key processes that
would be required to mechanistically simulate certain effects. We are aware of no models that represent Ni toxicity, for
instance. Including these effects, as well as a capacity to simulate secondary interactions, such as ligand complexation of
metals in OAE feedstocks, will require significant investment in empirical experimentation to understand essential rate
processes and physiological responses.

Shortcomings in the capacity of models to represent physiological responses to OAE is an important consideration for the
ability of models to faithfully represent ecological impacts. Notably, electrochemical OAE techniques present a simpler set of
processes to consider than using crushed-rock feedstocks, where ancillary constituents and physical dynamics come into play.
For electrochemical OAE, the most likely biological feedback to consider relates to the impacts of changing carbonate
chemistry on biogenic rates of calcification or phytoplankton growth rates (Paul & Bach, 2020). It is also possible that carbon
limitation of phytoplankton growth (Paul and Bach, 2020; Riebesell et al., 1993) may also be important. Empirical research
exploring physiological sensitivities should be used to develop prioritizations of key model processes comprising early targets





for implementation. Model documentations should use consistent stoichiometric relations to link alkalinity changes to those of nutrients and carbon (Paulmier et al., 2009) and state the assumptions made about carbonate formation and dissolution.

### 2.2.3 Representing sediment-water exchanges

The exchange of solutes between the sediments and overlying water influences ocean chemistry, including the properties of the carbonate system (Burdige 2007). Depending on location and time scale, OAE may affect these exchanges and should be
appropriately considered in models. Sediments influence the marine carbonate system primarily through the remineralization of organic matter, which returns DIC to overlying water (and alkalinity if this remineralization occurs anaerobically), and the dissolution of biogenic silicate or carbonate minerals. $CaCO_3$ is of particular importance as its dissolution releases alkalinity, while its burial is an alkalinity sink, and the balance between the two is a key control on the whole ocean alkalinity balance over timescales approaching 104 years (Middelburg et al. 2020). Furthermore, remineralization and other microbial
metabolisms, such as "cable bacteria," can significantly lower pore water pH by several pH units below seawater values (Meysman and Montserrat 2017). This can drive dissolution of $CaCO_3$ and alkalinity generation in the sediments, even in shallow waters when the overlying water is supersaturated (Rau et al. 2012).

Representing these processes in coastal and shelf sediments (< 200 m) is challenging. Shallow water depths and high productivity result in a significant delivery of organic matter to the sediments that is much larger than in the deep ocean. As a
result, the relative importance of sediments in organic matter remineralization is larger and production of alkalinity by anaerobic metabolisms is more important in these shallow sediments than in the deep ocean (Seitzinger et al. 2006, Jahnke 2010, Huettel et al. 2014, Chua et al. 2022). In addition, these environments are dynamic with organic supply and bottom water conditions varying on tidal, seasonal, and interannual timescales. Accounting for the exchange between sediments and overlying water and its variability on tidal, seasonal, and interannual timescales will likely be necessary in regional and global
biogeochemical models that aim to simulate alkalinity cycling in coastal and shelf seas, even for relatively short simulation durations of months to years.

The choice of approach to modeling sediments may depend on the sediment type. For example, the mechanisms transporting solutes across the sediment-water interface can be divided into two categories depending on the sediment's grain size. In coarse sediments—i.e. permeable sands—pressure gradients drive flow through the seabed replenishing sediment oxygen content
(Huettel et al. 2014). Organic carbon stores are low and remineralization was long thought to be primarily aerobic. However, evidence has emerged relatively recently that anaerobic remineralization in sandy sediments is more important than originally thought (Chua et al. 2022 and references therein). Idealized models that represent the three-dimensional sediment structure illustrate the importance of turbulence and oscillatory flows in permeable sediments (see Box 2 in Chua et al. 2022). These models are highly localized and computationally demanding, prohibiting their coupling with ocean biogeochemical models.
Thus, permeable sediments are currently not well represented in regional or global ocean biogeochemical models.





In cohesive, fine-grained sediments with low permeability, i.e., muds, transport is limited by diffusion or faunal mediated mixing and exchange processes, i.e., bioirrigation or bioturbation (Meysman, et al. 2006, Aller 2001). In these environments, detailed multicomponent reactive-transport models of sediment biogeochemistry – so called diagenetic models – can reproduce carbon remineralization rates partitioned between aerobic and anaerobic pathways, precipitation/dissolution reactions between

sediment grains and porewaters, and the transport of solutes across the sediment-water interface (Boudreau 1997, Middelburg et al., 2020). These mechanistic models will be useful for detailed investigations into how perturbations of the carbonate system in seawater overlying the sediments affect their biogeochemistry and for addressing questions about the potential influence of particulate alkalinity feedstocks settling to the seafloor (Montserrat et al. 2017, Meysman and Montserrat 2017). However, typically these models are one-dimensional and applied to a few representative locations. Coupling fully explicit diagenetic

models to three-dimensional ocean biogeochemical models, while conceptually straightforward, is computationally prohibitive. Instead, depth-integrated sediment processes have been implemented as bottom boundary conditions (e.g., Moriarty et al. 2017, 2018, Laurent et al. 2016). For example, Laurent et al. (2016) used a diagenetic model in a "meta-modeling" approach to estimate bottom boundary nutrient fluxes for a regional scale biogeochemical model. By parameterizing the diagenetic model with detailed geochemical data (porewater profiles and nutrient fluxes) from a few individual locations,

then forcing it over a range of expected bottom water conditions, they developed empirical functions relating sediment fluxes to bottom water conditions that could be used to parameterize bottom boundary conditions in the water column model. A similar approach could be used in OAE models to parameterize how sediment biogeochemistry may alter alkalinity fluxes, for example, how redox sensitive processes, such as coupled nitrification-denitrification or sulfate reduction coupled to pyrite burial, both of which may produce alkalinity (Soetaert et al. 2007), may respond to changes in bottom water oxygen or organic

matter loading.

When considering the long-term storage of $CO_2$ in global-scale ESMs, the interactions between sediments and the deep ocean (> 1000 m bottom depth) may need to be considered. In this environment most organic matter remineralization occurs in the water column, and the small amount of organic matter reaching the seafloor is remineralized aerobically with little to no release of alkalinity. In this case, sediment remineralization can likely be either ignored or implemented as a reflective boundary

condition where the simulated POC flux to the seafloor is immediately returned as DIC and remineralized nutrients. However, the dissolution or preservation of $CaCO_3$ in deep sediments is critical to controlling deep water alkalinity and may be important in model simulations that aim to quantify OAE effects on the timescales associated with the large-scale global overturning circulation. $CaCO_3$ solubility increases with pressure and pH and eventually becomes undersaturated at depth. The depth at which sinking $CaCO_3$ balances its dissolution is referred to as the carbonate compensation depth (CCD). An increase in bottom

water $CO_3^{2-}$ or $CaCO_3$ deposition, will deepen the CCD, burying $CaCO_3$, trapping alkalinity, and lowering the alkalinity budget of the ocean. Conversely if $CaCO_3$ rain rate or $CO_3^{2-}$ concentration decreases, the CCD will shoal and previously buried $CaCO_3$ will dissolve releasing alkalinity to the deep ocean. CCD compensation therefore opposes any forcing of the deep ocean carbonate system and therefore dampens the rise of $CO_2$ in the atmosphere but will also counteract any potential OAE solution



(see Renforth and Henderson 2017 for a detailed explanation). Although most CaCO$_3$ dissolution occurs in the sediments,
there is no consensus as to the level of detail this needs to be represented in models. Some global models employed to
investigate large-scale OAE include calcium carbonate dynamics at the sediment surface (Ilyina et al. 2013) others disregard
this process (Keller et al. 2014).

Often global models will parameterize CaCO$_3$ burial as a function of saturation state, such an approach is effective for resolving
CCD dynamics over geological timescales (~10,000 y), but not over the century to millennial timescales of CCD readjustment.
Models that fully couple sediment diagenesis can resolve these dynamics (Gehlen et al. 2008), but the computational demand
can make them ineffective. One solution is the approach of Boudreau et al. (2010) and (2018). By suggesting that CaCO$_3$
dissolution dynamics are controlled by transport of dissolution products across the benthic boundary layer, they were able to
derive equations predicting CCD depth and CaCO$_3$ dissolution based on bottom water CO$_3^{2-}$ and CaCO$_3$ rain rate and avoiding
a detailed representation of the sediments. These equations, combined with model bathymetry, can parameterize sediment
CO$_3^{2-}$ flux as a boundary condition and suitably account for transient sediment CaCO$_3$ dissolution in large scale ESMs while
avoiding the computational demands of a fully coupled ocean circulation-diagenesis model.

### 2.2.4 Representing river and groundwater fluxes

Regional and global ocean biogeochemical models typically account for river inputs, including their contributions to alkalinity
and dissolved inorganic carbon. In most models this is done by specifying alkalinity and dissolved inorganic carbon
concentrations in imposed riverine freshwater fluxes, although accurate prescription of these concentrations can be
challenging. Typically, a combination of direct river measurements, where available, output from watershed models (e.g.,
Seitzinger et al. 2010), or extrapolations of coastal ocean measurements to a freshwater endmember (e.g., Rutherford et al.
2021) are used. Solute inputs from groundwater are typically ignored but could be important locally. In high-resolution coastal
domains near urban areas, sewage input may be an additional important source of carbon, nutrients and alkalinity.

It is important to note that land-based CDR applications may have an important effect on ocean alkalinity dynamics through
riverine and groundwater delivery of solutes. Terrestrial OAE equivalents broadly referred to as Enhanced Rock Weathering
(ERW) rely on the application of lime or pulverized silicate or carbonate rocks on land and in rivers. These strategies aim to
generate CO$_2$ uptake locally but yield a leaching flux of bicarbonate into freshwater systems and subsequent transport into the
coastal ocean. Field trials and some commercial applications are currently underway, most of them with the implicit or explicit
assumption that the enhanced delivery of alkalinity will generate a carbon removal in the ocean (Köhler et al., 2010; Taylor et
al., 2016; Bach et al., 2019). There is a critical need for coordinated efforts to improve quantification of background riverine
fluxes and establish initiatives to effectively track the solute additions from ERW.



### 2.2.5 Representing air-sea gas exchange

The calculation of air-sea gas exchange is necessary for the quantification of net carbon uptake from OAE in models.
Biogeochemical models typically represent this exchange using a bulk relationship that depends on the product of the gas transfer velocity and the effective air-sea concentration difference (Fairall et al., 2000). However, the gas transfer velocity remains highly uncertain and is sensitive to a collection of processes that vary across scales, including sea state, boundary layer turbulence, bubble dynamics, and concentrations of surfactants. The most widely used parameterizations of the gas transfer velocity use empirical fits to observations to construct a functional relation dependent on wind speed only, under the
premise that turbulence and bubbles (via the breaking of surface gravity waves) are predominantly determined by wind stress (Wanninkhof, 2014). This neglects processes that could be regionally important such as convection, modification by biological surfactants, rain and wave-current interactions, while vastly simplifying the effects of wave breaking and bubbles. Although different dependencies on wind speed have been proposed (quadratic, cubic, hybrid), parameterizing the gas transfer coefficient as a quadratic function of the 10-meter wind speed is the most common (Wanninkhof, 1992; Wanninkhof 2014). This
relationship is supported by direct measurements of air-sea flux at intermediate wind speeds (3-15 m/s), but at low wind speeds (< 3 m/s), non-wind effects can have an important impact on gas transfer. At high wind speeds (> 15 m/s), breaking waves and bubble injection enhance gas exchange for lower solubility gasses such as $CO_2$ (Bell et al., 2017). Therefore, quadratic fits tend to underestimate the gas exchange at low and high wind speeds (Bell et al., 2017).

More complex air-sea exchange parameterizations account for processes such as bubbles, near surface gradients and buoyancy
driven convection (e.g., Liang et al., 2013, Fairall et al., 2000), but they depend upon a wider range of input variables. Other considerations in estimating flux arise from the nonlinear dependence on these variables, for example wind speed, which can lead to underestimates when made using daily averages rather than hourly measurements (Bates & Merlivat, 2001).

Notably, the gas transfer velocity (kw) determines the kinetics of gas exchange, given a perturbation in surface ocean $p$CO$_2$ away from equilibrium. The timescale for $CO_2$ equilibration over the surface mixed layer can be fully quantified using the
following expression,

$$\tau_{gas-ex} = \left(\frac{\partial CO2}{\partial DIC}\right)^{-1}\left(\frac{h}{k_w}\right), \tag{1}$$

where $h$ is the depth of the surface mixed layer and the partial derivative $\partial CO_2/\partial DIC$ captures the thermodynamic state of the carbon system chemistry in seawater, specifically with respect to the amount that dissolved $CO_2$ changes per unit change in DIC (Sarmiento and Gruber, 2006). This property is related to the buffer capacity and varies in roughly linear proportion to
the carbonate ion concentration. The magnitude of $\left(\frac{\partial CO2}{\partial DIC}\right)^{-1}$ is typically about 20, which explains why the equilibration timescale for $CO_2$ is so long. The contribution of uncertainty in the gas exchange velocity to overall uncertainty in carbon uptake from OAE deployments will depend in part on the circulation regime involved. For example, in situations where





alkalinity-enhanced water parcels are retained at the surface for timescales that are significantly longer than $\tau_{gas\text{-}ex}$, full equilibration will occur and the impact of uncertainty in the gas exchange velocity will have limited influence on the overall

uncertainty.

Even though OAE-induced additional air-sea $CO_2$ fluxes will, even in hypothetical massive deployments, amount to at most a few Gt $CO_2$/yr, which is typically not more than a percent of the atmospheric $CO_2$ inventory, this subtle difference in the treatment of the atmospheric boundary condition can be significant. Using prescribed atmospheric $pCO_2$ that is unresponsive to marine CDR-induced air-sea $CO_2$ fluxes has been shown to overestimate oceanic $CO_2$ uptake by 2%, 25%, 100% and more

than 500% on annual, decadal, centennial and millennial timescales, respectively (Oschlies, 2009). Simulations with prescribed atmospheric $pCO_2$ need to keep such systematic biases into account.

### 2.3 Model development needs for OAE research

While there is already substantial capacity for simulating ocean biogeochemical dynamics at global to regional scales, the discussion above implicates several areas where additional efforts are required to fully establish a modeling capability suitable

for supporting OAE. These fall into four primary areas: (1) supporting multi-scale simulations with sufficiently high-fidelity flow fields; (2) faithfully simulating the near-field dynamics associated with alkalinity addition; (3) capturing feedbacks to OAE owing to biological and geochemical responses; and (4) identifying whether there are reduced-complexity modeling approaches that might provide sufficiently robust estimates of the net effects of OAE.

As elucidated above, a primary consideration related to capturing OAE impacts is the fidelity of the simulated flow. Notably,

OAE presents a somewhat novel use case requiring an effective multi-scale modeling capability. A conceptually straightforward path to improving the representation of ocean circulation and mixing is to increase the resolution of the model grid. However, the computational demand of high-resolution simulations can only be met over more limited-area domains. Since the spatiotemporal footprint of OAE-related perturbations is likely to be large, there will be a need to represent large regions. An argument might be made, however, that the circulation in proximity of an OAE site is most important to capture with high-fidelity

with high-fidelity. This can be achieved with two-way nested regional models as described in see Section 2.1.2, but will require further development to couple in the nearfield models described in Section 2.1.1. Native grid-refinement, is another approach that may be pursued to effectively support OAE research.

The second area of model development relates to the requirement of faithfully representing the dynamics associated with alkalinity addition. Regional to global scales are the most relevant for simulating the air-to-sea exchange of $CO_2$ ensuing from

OAE. It is important, however, to ensure that local processes affecting the mass fluxes and initial dispersal of alkalinity are handled appropriately. As illustrated above, DNS or LES simulations (section 2.1.1) can be leveraged to develop parameterizations for larger-scale models, including for crushed-rock feedstocks where particle dynamics may be important or techniques involving alkalinity enhanced streams entering the ocean from outfall pipes. In addition to process fidelity, there





are also numerical constraints to consider. For example, advection schemes used in most ocean general circulation models struggle to represent sharp gradients; large mass fluxes of alkalinity into single model grid-points are likely to cause advection errors that may contaminate aspects of the model solutions making interpretation difficult. More specifically, conservative advection schemes can be characterized in terms of their accuracy, monotonicity (i.e., ability to preserve sign), and linearity (i.e., ability to preserve additivity) and there are always tradeoffs to make between these properties. Research may be required to determine which schemes are best suited to the particular challenges associated with representing the advection of OAE signals.

The third area of model development relates to our capacity to fully capture the range of biogeochemical feedback associated with OAE. The class of processes to consider here is potentially large and many have been touched on in section 2.2.1 to 2.2.3. Precipitation dynamics, specific elemental components of alkalinity, biogenic responses mediated by physiological or ecological sensitivities, impacts and processes controlling the cycling of ancillary constituents, and accurate sediment-water exchange are all areas that merit consideration. Further efforts are required to understand and prioritize these areas of potential development and, notably, their relative importance is likely to be regionally dependent.

Finally, it is important that models be tailored to address specific questions of relevance. In this context, it may be important to consider how much model complexity is required to capture the effects of perturbations, seeking parsimonious representations that are well-supported by empirical constraints and invoking wherever possible a separation of concerns so as to isolate factors contributing to uncertainty. For example, there are several near-field considerations that might be addressed using a combination of local observations and ultra-high-resolution modeling tools to generate estimates of alkalinity mass fluxes that are subsequently imposed as forcing in regional- to global-scale models. Another key question is how important it is to comprehensively simulate the mean state to faithfully capture the response to OAE perturbations. For example, if it can be documented that biological feedbacks to OAE are of negligible concern, the core target for simulating OAE effects may be to capture the cumulative integral of air-sea $CO_2$ exchange associated with the induced surface ocean $pCO_2$ anomaly. The mean state of the seawater carbon system is relevant here as the background DIC and alkalinity fields determine the $pCO_2$ response per unit addition of alkalinity, but fully prognostic calculations of nutrient cycling may not be necessary.

## 3 Model validation and integration with observations

Whether a model is useful for OAE research depends on how accurately it represents the physical, chemical, and biological processes that are relevant to the specific research question to be addressed. Model validation, the evaluation of a model's performance, and estimation of uncertainties in model output should thus be integral parts of model implementation and application. It is important to note that any model, even after best efforts have been made to improve formulations and conduct the most thorough validation, will deviate from reality. Any model is, by definition, a simplification of the real world and thus its output will be subject to uncertainties. Deviations of the model state from the real world can be reduced by applying



statistical techniques, collectively referred to as Data Assimilation (DA) methods, that combine models with observations and yield the best possible estimates. The steps typically involved in model implementation and validation, and possible integration with observations through data assimilation are shown in Figure 4. In this section, we summarize the most important observation needs for model validation (Section 3.1), briefly describe typical metrics for model validation and articulate a reasonable minimum criterion (Section 3.2), give a high-level explanation of approaches for the formal statistical combination

of models with observations through parameter optimization and state estimation (Section 3.3), and describe approaches for the specification of uncertainty in model outputs (Section 3.4).

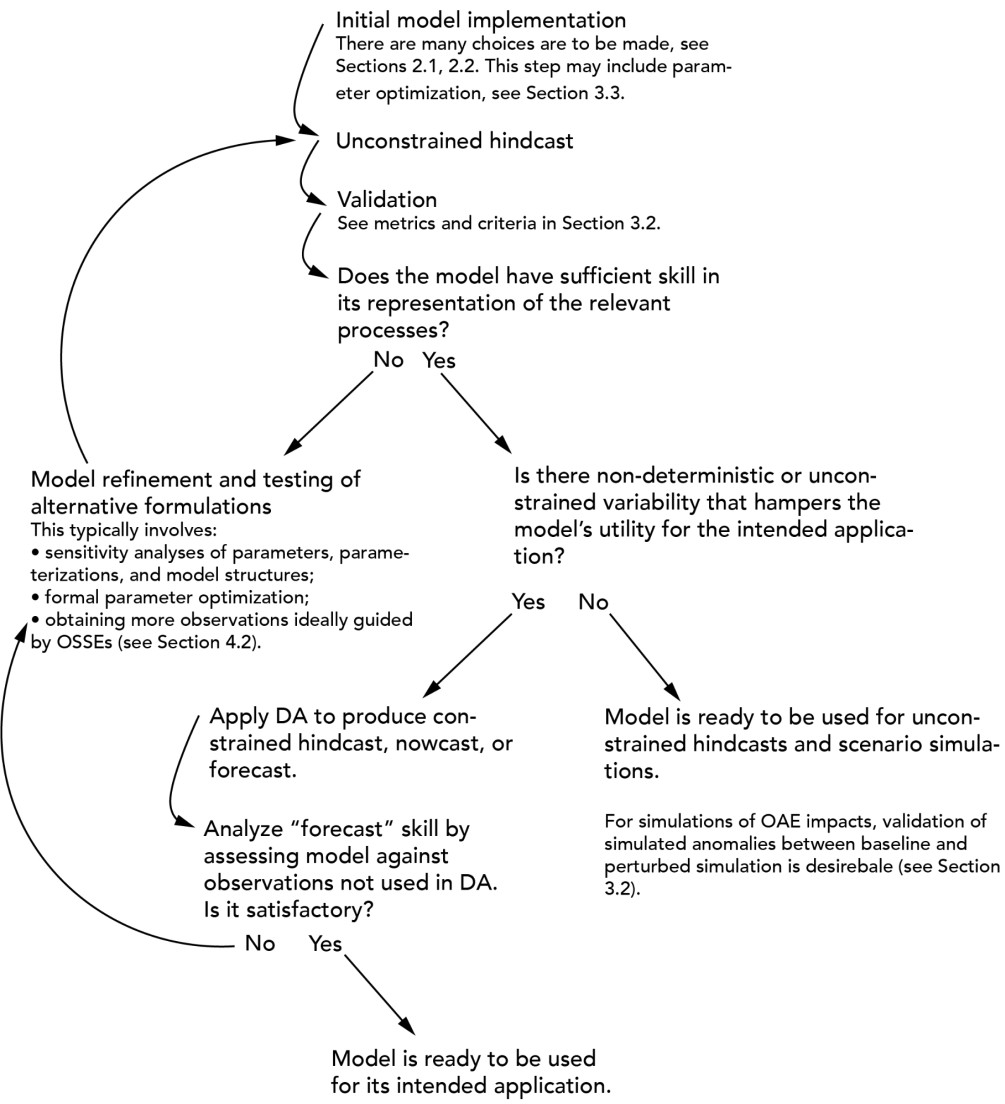

**Figure 4:** Typical steps in model implementation and validation.



## 3.1 Observation types for validation

Two fundamental requirements for models to be useful in the context of OAE research are high-fidelity representations of physical transport due to advection and mixing, and of biogeochemical effects of OAE, most importantly changes in the inorganic carbon properties.

Observations for validation of the simulated physical transport of alkalized waters include temperature and salinity distributions, direct measurements of currents, surface drifter trajectories, sea surface height observations from satellite altimetry, and estimates of geostrophic flow derived from the latter. Additional metrics relevant for assessing the fidelity of the large-scale overturning circulation in global models include combinations of biogeochemical concentration and transient tracers. For example, oxygen can be useful for identifying large-scale transport pathways, even though it convolutes dynamical and biological information. Particularly valuable for assessing large-scale ocean transport on the timescales relevant for OAE are abiotic transient tracers such as such as chlorofluorocarbons (CFCs), sulfur hexafluoride ($SF_6$), and possibly the isotopes $^{39}Ar$ and $^{14}C$. Observational approaches for validation at regional scales include explicit tracer studies for documenting dispersion properties using rhodamine dye or $SF_6$.

In addition to the dynamics of the flow, model validation for OAE research requires the assessment of the fidelity of simulated carbonate chemistry variables (e.g., alkalinity, total dissolved inorganic carbon or DIC, pH, partial pressure of $CO_2$ [$pCO_2$]) and salinity and temperature, which are used to calculate the 13 thermodynamic equilibrium constants and conservative chemical species needed to constrain seawater acid-base chemistry in oxygenated seawater. Depending on the OAE approach and the model application, assessment may also require observed macronutrient (e.g., nitrate, silicate, or phosphate), micronutrient (e.g., Fe), and contaminant (e.g., Ni, and Cr) measurements; bulk seawater properties related to biogeochemical cycling (e.g., dissolved organic carbon content [DOC], particulate inorganic carbon [PIC], chlorophyll fluorescence); and biogeochemical rates and fluxes (e.g., net community calcification).

It is not always feasible to obtain the ideal carbonate system observations for model validation. Temperature and salinity can be measured reliably across all ocean depths and, with greater uncertainty and only at the ocean surface, remotely from satellites. The technical capacity for seawater pH measurements is evolving rapidly and sensors and systems now exist for pH measurements across nearly all depths, though the depth-capable systems require regular recalibration (e.g., Maurer et al., 2021). Similarly, there are numerous ways to observe surface ocean $pCO_2$ using a variety of crewed, autonomous, and fixed-location platforms (e.g., ship-based, Saildrone, and moored systems). However, interior-ocean $pCO_2$ observations remain challenging to obtain due to the need for calibration gasses and a gas-water interface. Alkalinity titrations are predominantly performed on discrete bottle samples collected by hand, though autonomous titration systems are under development that enable in situ surface time series measurements (Shangguan et al., 2022). Microfluidic in situ alkalinity titrators are also under development that consume less reagent per sample but currently show higher uncertainties than discrete samples (Sonnichsen




et al. 2023). Solid state titrators that generate acid titrant in situ show promise for surface and subsurface alkalinity titrations, but these sensors are still undergoing development and validation (Briggs et al., 2017). DIC observations combine the limitations of current measurement systems for both the $p$CO$_2$ and alkalinity, and there are only a handful of automated DIC titration systems rated for surface ocean measurements (e.g., Fassbender et al. 2015; Wang et al. 2015; Ringham 2022).

Theoretically, measurement of two of the carbonate system parameters in combination with temperature and salinity and some additional assumptions allows calculation of the other carbonate system parameters in seawater. Unfortunately, the pair of $p$CO$_2$ and pH, which are the most accessible to autonomous measurement among the carbonate system parameters, provide nearly identical information about the system. Thus, the results of the calculations that use this pair have higher uncertainties than other combinations (Dickson and Riley 1979; Millero 2007; Cullison Gray et al. 2011; McLaughlin et al. 2015; Raimondi

et al. 2019) and are therefore not ideal as a pair for model validation.

### 3.2 Validation metrics and approach

Validation relies on comparing the model output to observations, often in an iterative loop where the evaluation of a hindcast simulation is followed by model refinements followed in turn by a new hindcast and re-evaluation (Figure 4, Rothstein et al. 2006). Several evaluation metrics are commonly used (see Box 3 in Fennel et al. 2022). The three most common are the root-

mean-square error (RMSE), the bias, and the correlation coefficient. All three are relative measures without any objective criterion that indicates which range of values is acceptable or unacceptable. In contrast, the Z-scores, which consider variability within the observational data set, and the so-called model efficiency or model skill, which quantifies whether the model outperforms an observational climatology are two metrics with built-in criteria as to whether a model's performance is acceptable or not (Fennel et al. 2022). Since no single metric provides a complete picture of a model's skill, multiple

complementary metrics should always be used in combination (Stow et al. 2009). Furthermore, different points in space and time, and a breadth of variable types should be part of any comprehensive validation because a model may provide accurate estimates for some variables, locations, or times but perform poorly for others (Doney et al. 2009).

For OAE research, validation can be considered as a two-step challenge. First, it is necessary to validate unperturbed model baselines to gain confidence that the natural variability is represented appropriately and to quantify model uncertainties. One

should compare model-simulated spatial fields and time-series at strategic locations with appropriate observations to assess the model's skill at representing mean distributions as well as the variability for carbonate chemistry measurements and other relevant properties using several of the complementary quantitative metrics listed above. A model could be considered as sufficiently validated when mean distributions, their seasonal variability, and the timing and magnitude of events (e.g., blooms, physical disturbances) are accurately represented. As described in Section 3.1, insufficient availability of observational

constraints on carbonate system parameters presents a major challenge in this regard. In models applied for OAE research, it is particularly important to assess whether they realistically capture the distributions and variability of seawater properties that





govern sensitivity of the seawater carbonate system; recent work by Hinrichs et al. (2023) shows that the current representation of alkalinity in state-of-the-art models requires improvements.

The second, even more difficult step is to test whether a model accurately represents alkalinity additions. OAE-related
modeling studies thus far have relied on models that are validated only for baseline conditions. These are useful as sensitivity studies. However, validation of a model's ability to accurately represent the perturbations of an alkalinity addition is ultimately needed to address OAE science questions around environmental impacts and MRV. It is likely that the metrics described above for baseline validation are not suitable for this task. Validation should focus on quantifying whether the model accurately captures the anomalies created by OAE. This requires consideration of the spatial footprint and temporal evolution of
perturbations and ideally a close integration of experimental, observational, and modeling efforts. For example, a model that is deemed skillful after baseline validation can be used to estimate the appropriate dosage of alkalinity additions, thus ensuring a measurable signal, and guide the observational strategy; subsequent validation may indicate model shortcomings that were not obvious in the baseline validation (e.g., diverging dissipation rates between model and field observations) and prompt model refinement in an iterative loop of model validation, improvement, and renewed experimental assessment (Figure 4).

It is important to note that even with repeated steps of validation and model improvement, there is going to be a limit to the degree of realism that can be achieved with any model. Any model simulation will be prone to errors and uncertainties. Sources of error include inaccuracies in model inputs, numerical approximation schemes, insufficient process understanding, and inaccurate model parameters and parameterizations.

### 3.3 Data assimilation

Data assimilation (DA) is the process of improving the dynamical behavior of models by statistically combining them with observations. There are a variety of DA techniques that rely on different mathematical and statistical approaches (Carrassi et al. 2018). Originally developed for numerical weather prediction, DA has been successfully applied to ocean models, including biogeochemical models (Mattern et al. 2017, Cossarini et al. 2019, Ciavatta et al. 2018, Verdy and Mazloff 2017, Teruzzi et al. 2018, Fennel et al. 2019) but success critically depends on the information content of the available observations (Yu et al.
2018; Wang et al. 2020). While DA has been shown to yield large improvements in important parameters governing biogeochemical processes (Mattern et al. 2012, Schartau et al. 2017, Wang et al. 2020) and in model estimates of the physical and biogeochemical model state (Hu et al. 2012, Mattern et al. 2017, Ciavatta et al. 2018), it is only starting to be applied to carbonate system properties (Verdy and Mazloff 2017, Carroll et al. 2020, Turner et al. 2023, Figure 5).

Application of DA for ocean models is typically applied for one of two purposes: (1) to systematically optimize model
parameters, e.g., phytoplankton growth and nutrient uptake or rates of background dispersion, and (2) to estimate ocean state, e.g., distributions of temperature, phytoplankton biomass, alkalinity (see Fennel et al. 2022 for more details on the practical approaches and examples). The first purpose addresses systematic errors and biases in models and is useful when

systematically modifying and testing different model formulations while the second assumes an unbiased model and addresses
unresolved stochasticity, e.g., correcting the locations of mesoscale eddies and current meanders. Joint estimation of physical
and biogeochemical properties is common and can yield significant improvements for both types of properties (Yu et al. 2018).
Hybrid approaches combining parameter and state estimation have also been proposed (Kitagawa 1998, Mattern et al. 2012,
2014) but are less widely used.

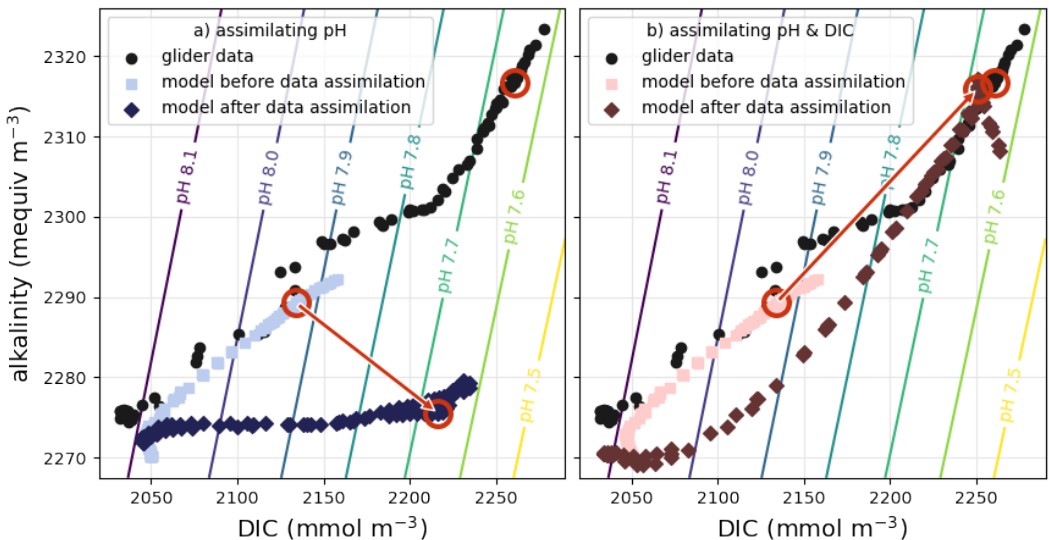

**Figure 5:** Example of a DA application for state estimation of carbonate system properties within a 3-dimensional model of
the California Current System. The symbols show glider data and model estimates at the measurement times and locations;
one specific data point and its associated model estimates are highlighted by red circles. Each data point consists of measured
pH alongside estimated alkalinity and DIC values (see Takeshita et al. (2021) for data source and details). In the model, pH is
a diagnostic variable and primarily dependent on the model's alkalinity and DIC estimates. (a) When only pH data is
assimilated, the model estimates are moved closer to the observed pH values by increments in alkalinity-DIC space that degrade
the model's alkalinity estimates. (b) The model state estimates improve considerably by assimilating data for DIC (or alkalinity;
not shown) together with the pH observations.

Successful application of DA critically requires sufficient observations either of the properties that the model parameters to be
estimated depend on or of the state variables that are being estimated. The most commonly used observation type in
biogeochemical DA applications is satellite-based ocean color observations (Mattern et al. 2017, Ciavatta et al. 2018, Teruzzi
et al. 2018) which are available at a relatively high temporal resolution and covering large areas of the surface ocean. While
these observations are useful for informing model estimates of properties directly linked to processes involving phytoplankton,
they provide little information on the carbonate system. Dynamical models are able to quantitatively constrain processes that



cannot be measured directly, by inferring them from observable properties. But only if the observations contain enough relevant information about the processes of interest. Hence, one of the biggest challenges facing the application of DA to models of the marine carbonate system, is the sparsity of observations of the marine carbonate system. Although measurements of DIC, pH, alkalinity, and $p$CO$_2$ are becoming more common (Chai et al. 2020), they are still sparse compared to what is typically required for DA applications. In this context, an additional challenge is the problem of underdetermination: If multiple processes of interest can cause a similar change in an observable property, then observing this property alone may not be enough information to constrain these processes and more observations are needed (see Fig. 4 and code examples in Fennel et al. 2022).

In situ measurements of the carbonate system are much more limited temporally and especially spatially than remote sensing observations from satellites, which are the backbone of physical (satellite sea surface temperature and sea surface height) and biogeochemical data assimilation (see above). Observations of pH, $p$CO$_2$, alkalinity, and DIC used to be limited to moorings and research cruises but have more recently been extended by automated observing systems, such as gliders, BGC-Argo floats and uncrewed surface vehicles (Bushinski et al. 2019). As new platforms are added to the observing system, DA techniques can help guide their optimal deployment (see Section 4.3 below). Furthermore, statistical and machine-learning approaches are being developed (e.g., Lohrenz et al. 2018, Bittig et al. 2018, in prep.) that may help overcome the undersampling of carbonate system properties and could feed directly into DA applications.

There is an important subtlety to the application of data-assimilative models when quantifying net CO$_2$ uptake due to OAE, which is highly relevant for MRV. When the net CO$_2$ uptake is quantified by calculating the difference between two simulations, one with and one without OAE (one of these is realistic, the other counterfactual), it is not appropriate to assimilate biogeochemical observations of properties affected by the alkalinity enhancement. The assimilation of alkalinity-related observations to constrain one of the simulations in the pair would eliminate the ability to make comparisons between the two. However, assimilation of observations that are unaffected by OAE (e.g., temperature, salinity, oxygen, etc.) can be applied to both simulations of the pair. Further research and method development are required to identify the best approaches for leverage DA in this context.

### 3.4 Uncertainty analysis

Model results should be paired with sound qualitative and quantitative uncertainty estimates, especially when used for practical decisions. Estimating the uncertainty of model simulations, however, is inherently difficult because typically one is most interested in simulation outputs for which observations are not available (e.g., unobserved or insufficiently observed properties or fluxes in the past, properties and fluxes in the future); hence, standard procedures and metrics for model validation (Section 3.2) are not helpful for this aspect. Uncertainty estimates could be based on extensive model parameter and configuration sensitivity studies and comparisons with models that include more realistic representations of uncertain or parameterized





processes. Furthermore, since specification of uncertainty is an integral part of DA, DA methodologies provide a useful framework for estimating uncertainty, especially ensemble-based methods.

Any DA application requires uncertainty specification of the observations that are assimilated and can provide uncertainty estimates of the results of the assimilation procedure. Specification of uncertainty in the input data is necessary to inform the DA machinery about how much weight and reach each data point or data type should have in influencing the outcome. The more realistic the uncertainties of the input data, the better the DA outcomes in terms of explanatory or predictive skill. It is important to note that better does not mean more precise, in this context. Overconfidence in the accuracy of assimilated
observations will lead to overfitting and a degradation of predictive skill. In the case of parameter optimization, the output of the assimilation exercise is a set of optimized parameters. The uncertainty of optimal parameters, referred to as *a posteriori* errors, is determined by a Hessian analysis of the cost function in combination with the uncertainty of the input parameters before optimization, the so-called *a priori* errors (Thacker et al. 1989, Fennel et al. 2001). In the case of ensemble-based state estimation, the ensemble spread of the reanalyzed model state provides a spatially and temporally resolved estimate of the
uncertainty of the reanalysis (Yu et al. 2018, Hu et al. 2012).

However, an important caveat is that subjectivity enters the uncertainty specification in all of these approaches. For example, in the case of parameter optimization the assumed *a priori* errors, their probability distributions, and the choice of the cost function are subjective and influence the *a posteriori* errors (but interestingly the values of the observations themselves do not). In the case of ensemble-based state estimation, the sources of uncertainty inherent in the model simulation have to be
specified and simulated by generating variations within a model ensemble. Sources of uncertainty include errors in atmospheric forcing and boundary conditions, model parameters, and structural uncertainty. Uncertainty in forcing and boundary conditions is often represented by perturbing the time of sampling, uncertainty in parameters is represented by sampling from a probability distribution (based on *a priori* assumptions about the uncertainty of each parameter), and the structural uncertainty is typically represented via brute-force inflation factors that amplify ensemble spread. Yu et al. (2019), Li et al. (2016), and Thacker et al.
(2012) provide examples where different sources of model uncertainty are accounted for. While the mechanics by which the model ensemble is generated and spreads over time is thus subjective, grossly inappropriate choices will lead to obviously wrong or degraded reanalyses. The success of a DA exercise, which is best judged by an evaluation of whether the predictive power of the model has improved, thus provides a useful reality check on whether the choices for specifying uncertainty were appropriate.

How can the framework for specifying and estimating uncertainty from model ensembles be applied in the context of OAE research? Two different cases should be considered here: 1) model applications where the absolute value of quantities matters for the research question to be addressed and thus the uncertainty of the simulated output, and 2) applications where information about the difference between a simulation with and without OAE is of interest and the uncertainty of this difference (e.g., the net $CO_2$ uptake and its uncertainty in the context of MRV). Examples of the first case include studies of the stability



of added alkalinity (i.e., simulation of runaway calcium carbonate precipitation) and studies about the exposure of planktonic and benthic communities to high pH. In this case, the ensemble framework described above can be applied with the caveat that the specification of all the relevant sources of uncertainty is by no means trivial and subjective to some degree.

The second case is highly relevant for MRV of OAE where one is interested in accurately quantifying the increase in seawater DIC due to OAE with well characterized uncertainty. In this case, one would use two simulations that are based on an identical

model set-up with only one difference, namely a source of alkalinity is applied to one (i.e., one of these two simulations is counterfactual or hypothetical, the other would typically be as realistic as possible). It may be tempting, and is conceptually straightforward, to apply the ensemble framework for each model of the pair and combine the resulting uncertainties via error propagation. However, in practice this would not provide meaningful estimates because there are sources of uncertainty that are unaffected by OAE (e.g., atmospheric forcing) and accounting for them may significantly overestimate uncertainty in the

estimated net $CO_2$ uptake. A more appropriate approach would be to construct an ensemble of model pairs that explicitly accounts for uncertainty related to the impacts of alkalinity addition. How to specify and simulate the sources of uncertainty directly resulting from OAE in practice remains an open research question.

## 4 Model experimentation

In this section we lay out general objectives for model experimentation in the context of OAE research and provide a short

historical view of how these model studies have evolved (Section 4.1) followed by specific recommendations for Observing System Simulation Experiments (Section 4.2) and model intercomparisons (Section 4.3).

### 4.1 General objectives of model experimentation

General objectives of OAE modeling include (1) gaining a better understanding of the biogeochemistry of OAE, including its effectiveness and side effects, (2) supporting experiments, field trials, or commercial deployments including through the

optimization of observing systems, (3) assessing global carbon-cycle and climate feedbacks, (4) understanding the role that OAE can play in climate mitigation efforts, and (5) supporting monitoring, reporting, and verification activities. At a conceptual level, model approaches for OAE can be classified as belonging into one of two groups: idealized or realistic. Idealized modeling approaches are typically driven by research questions of a fundamental nature and aim to develop or test hypotheses or provide improved process understanding while strongly simplifying a range of potentially complicating factors.

They are useful for illustrating cause-and-effect relationships and the range of plausible outcomes given strong assumptions. In contrast, realistic modeling approaches aim to include a broad range of contributing factors as accurately as possible and provide detailed hindcasts or predictions that, if the model has skill, can be used for a range of practical applications. In practice, the dividing line between idealized and realistic models is blurry. Of course, no model will ever simulate all aspects





of reality, hence even realistic simulations make many assumptions and are prone to errors from multiple sources. It can be
effective to apply idealized and realistic approaches in a complementary manner and iteratively.

It is illustrative to review briefly how modeling for OAE research has developed over the course of the last decade. Much of the early work on OAE used idealized models. Model simulations were designed to investigate whether the theoretical concept of OAE could remove large amounts of $CO_2$ on the global scale. Rather than trying to account for the technical and socio-economic constraints of OAE deployment, the model experiments were designed to investigate what would happen if surface
alkalinity was homogeneously increased by massive amounts via a constant addition rate over extremely large regions of the ocean, e.g., in all sea-ice free waters (Paquay and Zeebe, 2013; Keller et al., 2014; Ilyina et al., 2013; Köhler et al., 2010; Köhler et al., 2013). These simulated OAE deployments will never be realized, but the model results suggested that OAE can be viable as a CDR approach. A particular advantage of this idealized approach is that the effect of OAE was easy to detect against internal model variability, i.e., the signal to noise ratio is high. The next steps in modeling OAE have remained idealized
but have begun to introduce more constraints and better mechanistic or empirically derived components as experimental OAE date becomes available. Recently, modeling studies tailored to specific regions and modes of application have been conducted to support field trials or commercial deployment (Mongin et al. 2021, Wang et al. 2023). These applications of course have to be as realistic as possible. None of the modeling studies published to date have simulated an actual OAE field trial.

### 4.2 Recommendations for Observing System Simulation Experiments (OSSEs)

Observing system simulation experiments (OSSEs) use data-assimilative simulations to design new, or modify existing, observing systems such that deployments of observing assets, e.g., floats, gliders, moorings, or surface vehicles, is optimized. General overviews and best practices for OSSEs are provided by Halliwell et al. (2015) and Hoffman and Atlas (2016). Examples of applications to biogeochemical models include Ford (2021), Wang et al. (2020), and Denvil-Sommer et al. (2021). Their goal is to maximize the information gained from a new or modified observing system, while keeping the number of
required instruments, sensors, or deployments – and thereby cost and effort – low. OSSEs are especially valuable tools in the context of OAE research because the marine carbonate system is still undersampled, observing systems need to be designed and expanded, and new instruments deployed and configured (Boyd et al. 2023).

In practice, this is done with the help of a pair of two different models or model versions, also referred to as twin experiments, as follows. A simulation of one of the models is considered to be the "truth." This simulation is also referred to as the "nature
run" and synthetic observations are generated by subsampling this nature run. This subsampling can be repeated with different sampling schemes (e.g., different variable types, different numbers of profiles, transects, and/or fixed location time series, etc.) to represent different configurations of the observing system.  Finally, the synthetic observations are assimilated into the other model for which a non-assimilative simulation, the so-called "free run," is also available. The skill of this data-assimilative simulation, also referred to as the "forecast run," can be assessed against the free run using independent observations that are





also sampled from the nature run. In this way the impact of different sets of observations on the data-assimilative model can be measured and assessed.

While conceptually straightforward, care and consideration are required when setting up OSSEs. For example, the choice of the two model versions making up the twin is important. If the models chosen for the truth and forecast runs are versions of the same model implementation that were generated by perturbing initial, forcing or boundary conditions in one of them, the

method is referred to as the "identical twin" approach. If two different model types are used, they are "non-identical twins." The intermediate approach where the same model type is used but in different configurations (e.g., different physical parameterizations and/or spatial resolution) is referred to as fraternal twin. The identical twin approach has been more common in oceanic DA applications although atmospheric OSSEs have shown that it can provide biased impact assessments (Hoffman and Atlas, 2016) typically because the error growth rate between the truth and forecast runs is insufficient. A direct comparison

of the non-identical and identical twin approach for an ocean circulation model of the Gulf of Mexico has been conducted by Yu et al. (2019). In their assessment of the impacts of the existing observing system (consisting of satellites and Argo floats), the identical twin approach provided overly optimistic improvements in model skill after assimilation of data from some observing assets (specifically sea-surface height and temperature) but undervalued the contribution from temperature and salinity profiles. They concluded that skill assessments and OSSEs using the non-identical twin approach are more robust.

Similar concerns likely apply to OSSEs for biogeochemical properties, but this remains to be studied systematically.

### 4.3 Recommendations for intercomparison

A common approach to assessing model uncertainty are coordinated, multi-model studies, commonly called model intercomparison projects or MIPs. They can be used to explore the simulated range of model behaviors, to isolate the strengths and weaknesses of different models in a controlled setting, and to interpret, through idealized experiments, inter-model

differences (IPCC 2013). Carefully designed experiments can also offer a way to distinguish between errors particular to an individual model and those that might be more universal and should become priority targets for model improvement (IPCC 2013). These studies rely on common agreed-upon protocols for simulating certain processes and writing of diagnostic output to ensure that best practices are followed, and results are comparable (e.g., Griffies et al., 2016). The best-known model intercomparison project is probably the Coupled Model Intercomparison Project (CMIP, Eyring et al., 2016), which is currently

finishing up its 6th phase. Within CMIP6, the carbon dioxide removal intercomparison project (CDRMIP; Keller et al., 2018) is the first project to develop a model intercomparison experiment for ocean alkalinity enhancement. This and other MIP examples, including those conducted at smaller region scales (Wilcox et al., 2022), provide a blueprint for developing coordinated multi-model experiments.

The following key practices have proven useful in previous coordinated multi-model comparisons:



• Since broad participation is typically desired, the protocol should be straightforward for modeling groups to implement, otherwise few will have the resources to participate. In practice this means avoiding new implementations of complex code or requiring too many or too long simulations.

• If applicable, forcing data should be centrally prepared and provided to participants in a standardized way that enables easy modification or reformatting, if needed, for use with different models.

• Using common simulations that modeling groups are likely to have completed already, e.g., climate change scenarios, as control runs and experimental branching points is helpful for minimizing the number of additional required simulations.

• It is useful to establish common practices that facilitate the production and analysis of the model output, e.g., what should be archived and shared (Juckes et. al., 2020) and data standards governing the structure and required metadata

for model output (Pascoe et al., 2020).

• Shared software to standardize model output, such as the Climate Model Output Rewriter (CMOR; https://cmor.llnl.gov/) commonly used in CMIP, can be helpful.

• To maximize the use of model output, it should be made available for public download with digital object identifiers (DOIs). The Earth System Grid Federation (ESGF) is an example of such a system (Petrie et al., 2021).

• If applicable, preparing and providing quality-controlled observational datasets for model evaluation is useful for facilitating analytical efforts (Waliser et al., 2020).

• Coordinating the analysis is helpful to avoid duplicative efforts and ensure consistent application of evaluation metrics.

• Finally, the design of a coordinated multi-model experiment and all its procedures should be well documented in

publications or permanently archived protocols.

It is advisable to test the multi-model experiment with a small subset of models, before inviting a large number of participants. Furthermore, it is worth remembering that the science questions must be appropriate. MIPs require much effort and not every science question needs a MIP to be answered.

**5 Summary**

A range of modeling tools and analysis methods are available for OAE research to address questions from micro- to global scales; however, each of these tools and methods has limitations and caveats that model users and users of model-generated outputs need to be aware of. Furthermore, this new field of research poses questions and challenges that current tools were not designed to address, necessitating further development.



A common objective of all modeling approaches described in this chapter is to simulate the spatio-temporal evolution of carbon
chemistry properties in seawater by accounting for the physical, chemical, and biological processes that determine this
evolution. Idealized models, which neglect some aspects of reality in the interest of simplicity and clarity of assumptions, have
long been used to test basic questions about OAE. As research questions are becoming more focussed on the practical aspects,
feasibility, and ecosystem impacts of OAE, more realistic models are increasingly desirable. A skillful realistic model can
provide spatial and temporal context for observations, including estimates of properties and fluxes not directly observed. Such
model will include parameterizations of the relevant processes for the research objective to be addressed and will be
constrained by observations that contain sufficient meaningful information. However, model formulations of several properties
and processes relevant to OAE research remain uncertain or highly simplified. For example, presently used model
representations of alkalinity in seawater are likely inadequate and may require explicit representation of at least some of the
multiple biotic and abiotic sources and sinks of alkalinity; the mechanisms and triggers for spontaneous calcium carbonate
precipitation are only beginning to be described and not yet represented in models; and the impacts of pH perturbations on
plankton diversity and trophic interactions remain an active area of study an unaccounted in biogeochemical models.
Furthermore, it is difficult to obtain solid constraints on the seawater carbonate system, especially in sufficient spatial and
temporal resolution for robust model validation and DA. Theoretically, knowledge of two of the carbonate system parameters
allows calculation of the others, but unfortunately $p\mathrm{CO_2}$ and pH, the pair most accessible to autonomous measurement, results
in high uncertainties.

One inherent challenge to OAE research is the multiscale nature of many of the relevant questions. Different modelling tools
are available for different spatial scales. While some research questions may fall neatly within the limited spatial range of a
particular model, many do not and require a bridging of scales that could be accomplished via new parameterizations yet to be
developed or dynamic coupling of different modeling tools. It is important to emphasize that models have to be tailored to the
questions they are meant to address. This means considering what level of model complexity is required and seeking
parsimonious representations that are well-supported by empirical constraints.

It is important to note that even after thorough validation, any model simulation will be prone to errors and uncertainties due
to inaccuracies in model inputs, structural uncertainty due to numerical approximation schemes and insufficient process
understanding or representation, and inaccurate model parameters and parameterizations. Deviations between models and
reality can be reduced by DA, which is typically applied either to systematically optimize model parameters or to produce
optimal estimates of the ocean state. Optimization of model parameters addresses systematic model errors and biases; it is
useful for systematic testing of different model formulations during model design. State estimation assumes an unbiased model
and addresses unresolved stochasticity, thus leading to model states that are in better agreement with the observed ocean state.
However, successful application of DA critically requires sufficient observations. Currently, the biggest impediment to
implementing data-assimilative models for OAE research is the sparsity of carbonate system observations. OSSEs, data-



assimilative simulations that inform how to place observing assets most effectively, will prove useful in this context. It should also be noted that assimilation of carbonate system parameters is not appropriate when models are applied for MRV.

Uncertainty analysis is a necessary component of any quantitative research and will be an essential deliverable for effective approaches to MRV. Ensemble-based DA methodologies provide a useful framework for estimating uncertainty. Consideration

of this framework illustrates the "law of conservation of difficulty" applies here. Quantitative assumptions about the uncertainty distributions of input data and input parameters, and of structural uncertainties inherent in the model are required to obtain an uncertainty estimate of the model output, in other words, difficult assumptions about errors have to made somewhere. A common approach to assessing model uncertainty is by coordinated, multi-model intercomparison. Such studies can be used to explore the range of simulated behaviors and the strengths and weaknesses of different models and, by

elucidating inter-model differences, they can offer guidance on priority targets for model improvement.

**Competing interests**

KF, AL, and CA are collaborating with Planetary Technology, a climate-tech company, and Pro-Oceanus Systems Inc., an ocean technology company, as part of an NSERC Alliance Missions project focussed on OAE; none of the partners have made direct financial contributions to the project. RM and JO are collaborating with Planetary Technologies, and JO is partially

supported by Planetary Technologies via an NSERC Alliance grant. ML is the Executive Director of [C]Worthy, LLC, which is a non-profit research organization focused on building tools to support MRV for ocean CDR.

**Acknowledgements**

This is a contribution to the "Guide for Best Practices on Ocean Alkalinity Enhancement Research." The ClimateWorks Foundation and the Prince Albert II of Monaco Foundation supported the participation of KF at the lead authors' meeting in

January 2023 at the Villefranche Oceanographic Laboratory and covered the page charges for this article. KF, AL, RM, and JO acknowledge funding by NSERC's Discovery and Alliance Programs. KF, DK, AL, and AO are supported by the Ocean Alk-Align project funded by Carbon to Sea. RM is supported by the Canada Research Chairs Program. JPM is supported by the Simons Collaboration on Computational Biogeochemical Modeling of Marine Ecosystems (grant ID: 459949FY22) and the NOAA Ocean Acidification Program (grant ID: NA19OAR0170357). DBW is supported by the NASA Earth science New

Investigator Program. Jessica Oberlander's assistance in compiling the bibliography is gratefully acknowledged.

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
