# Peer review of "Modeling considerations for research on Ocean Alkalinity Enhancement"

_State of the Planet, 2023_

## Author Comment (AC2)

**Reviewer 3 (Steve Rackley):**

I feel honoured to be asked to review the work of this distinguished group, and I hope that some at least of the following comments and suggestions—coming as they do from a non-oceanographer—will be helpful.

**Response:** We appreciate the careful and thorough review and are grateful for the kind words.

1. Line 51: I would argue that MRV is primarily a *deployment* rather than a *research* challenge, and that "skillful and fit-for-purpose models" will be **essential** in meeting this challenge, rather than merely "valuable".

**Response:** We agree that MRV is primarily a deployment challenge but feel that the question of how MRV can be done most reliably and effectively is an ongoing research challenge. We happily replace "valuable" with "essential." The sentence now reads (new text in bold):

*"Skillful and fit-for-purpose models will **be essential** prove valuable for addressing many OAE research questions including the MRV challenge, assessment of environmental impacts, and interpretation of natural analogs."*

2. Line 66: It would be helpful to expand on what is "sufficient", or how one would go about establishing that.

**Response:** What is sufficient depends on the application and research question to be addressed. For details on methods to establish that, we refer to section 4.2 about OSSEs. We have revised the sentence as follows:

*"Applications of realistic models rely on them being skillful and accurate, requiring that they include parameterizations of the relevant processes, and that they are constrained by observations that contain sufficient meaningful information **(what is sufficient depends on the application and research question)**."*

3. Line 77: Is the conflation of "scenarios" and "counterfactuals" standard nomenclature in this field? I am used to a different definition of scenarios, as not being limited to counterfactual cases. Where do simple sensitivities—essential in uncertainty analysis—fit into these four general types?

**Response:** We agree that counterfactuals are just a subset of scenario simulations and no, there is no standard nomenclature yet. The field is just emerging. We have changed this text as follows:

*"Scenarios, are unconstrained hindcasts or forecasts where one or more aspects of the model is systematically perturbed to assess the effect of the perturbation, for example, in paired simulations with and without OAE, one **would be the realistic case and the other a scenario (also referred to as counterfactual in this specific case)**. These can be used to explore even very unlikely situations, which is often required in comprehensive uncertainty and risk assessment."*

4. Line 84-85: "especially when considering" seems superfluous, as successful model implementation will be even more of a challenge for eventual OAE deployment. Also, modelling of OAE field trials involving small alkalinity additions may still involve large spatial and temporal scales, depending on local circulation and given the long gas exchange timescale.

**Response:** Agree. We have removed that phrase.

5. Line 115: Suggest to replace "mCDR" with "CDR", since terrestrial methods will also affect Earth system feedbacks.

**Response:** Agree. Changed mCDR to CDR.

6. Line 176: Might be useful to mention Lagrangian methods as also being relevant to modelling the physical dynamics of particles.

**Response:** Agree. We have added a sentence mentioning that this as a useful approach for this problem (new text in bold):

*"The least understood but potentially dominant source of uncertainty pertains to the representation of the microscale biological, chemical, and physical dynamics of particles, which is an active area of experimental and observational investigation (Subhas et al. 2022, Fuhr et al. 2022, Hartmann et al. 2023). **While the explicit multiphase modeling of the particles themselves is computationally costly, an approach wherein the parametrized evolution of inertia-less Lagrangian particles are simulated may provide a fruitful middle ground, providing a mechanism to realistically determine the alkalinity release field associated with the advection, mixing, sinking and dissolution of reactive mineral particles.**"*

7. Lines 246: "significant" is superfluous here, as even small-scale field trials are subject to the slow air-sea gas exchange. See note 4.

**Response:** Agree. We have removed "significant."

8. Line 248: I think it is incorrect to state that transportation distance is not relevant in the case of alkalinity "added to seawater that is oversaturated in $CO_2$". Such waters can also be transported large distances before equilibration (same gas exchange timescale applies!), and the seawater conditions at the point of

equilibration (not at the point of alkalinity addition) are what determine OAE efficiency.

**Response:** Agree that the same time scales for gas exchange apply in the undersaturated as in the oversaturated case. We have removed the first part of this sentence. It now reads as follows:

"*Unless equilibrated before the addition or added to seawater that is oversaturated in CO2,* *A*lkalinity-enhanced waters can be transported far away from injection sites before equilibration is complete (He and Tyka 2023)"

9. Line 295: "wastewater treatment plants" could also be mentioned in lines 290-293 in relation to reactive mineral addition (*Planetary*'s approach).

**Response:** Agree. Added "or through coastal outfalls (e.g., from wastewater-treatment or power plants)."

10. Line 313: "precludes" is perhaps too strong. Regional models may be sufficient if they fully contain the area of ocean circulation prior to subduction.

**Response:** Agree. We again removed "significant" in front of OAE deployments and removed the sentence that started with "This precludes…"

11. Lines 364-367: I do not think that **local** variations in relation to carbonate chemistry equilibrium coefficients impact eventual OAE efficiency. The relevant equilibrium coefficients are those at the point of air-sea gas equilibration or, more strictly, at the point of eventual subduction of the DIC enhanced waters, in the case that there are significant changes in seawater conditions between equilibration and subduction.

**Response:** This may be true for OAE deployment, which is why we use weak language here and say that "it may be important." We feel this is a relevant research topic.

12. Line 383: Challenging indeed, but perhaps not required for deployment modelling since, as suggested in line 542, there are likely reduced-complexity approaches that will be sufficient.

**Response:** Agree that this may not be required for routine deployment, but routine deployment should only proceed if it is found to be safe for ecosystems. Establishing the latter requires research and more complicated models.

13. Line 463: I guess this should be "and *decreasing* pH"?

**Response:** Yes, of course. Now corrected to say that CaCO$_3$ solubility increases with decreasing pH.

14. Line 496: Does this really stand out as "critical" among the many areas needing improvement?

**Response:** Agree. Removed the word "critical."

15. Lines 535-536: The concern regarding prescribed atmospheric pCO$_2$ is really only relevant in the case of conceptual global studies. For actual OAE deployments, hindcast modelling with the actual pCO$_2$ history will be required for the quantification of removals.

**Response:** Agree. We don't see any contradiction with what is written in the manuscript.

16. Lines 575-577: Re. "ultra-high-resolution modeling tools" if might be of interest to note (as a "personal communication"?) that *Planetary* is developing a Lagrangian particle tracking approach to determine the "alkalinity release field" generated by the advection, sinking, and dissolution of reactive mineral particles with a generic size distribution, which will then be imposed on a regional scale Eulerian model.

**Response:** We're reluctant to endorse specific approaches with attribution given the general nature of section 2.3 "Model development needs for OAE research," but agree that it is useful to mention Lagrangian methods and have done so now. Please see response to point 6 above.

17. Lines 580-581: "The mean state ...". See points 4 and 11 above.

**Response:** Please see responses above.

18. Lines 650-652: "different points ..." seems a bit too indeterminate, given that some points in space and time are far more relevant than others! (points 4 and 11 again).

**Response:** Please see responses above.

19. Line 910: While the pCO$_2$-pH pair results in high**er** parameter uncertainties, I don't think it is correct to characterize this as a "high" uncertainty in the overall context. (See e.g. CarbonPlan's Verification Framework, which characterizes carbonate system uncertainty as Low (1 to 5% impact)

https://carbonplan.org/research/cdr-verification/ocean-alkalinity-enhancement-mineral  Component 2; Mineral dissolution).

**Response:** We're making a different point here, namely that some pairs of the 6 carbonate system parameters provide more certain, or uncertain, estimates than others. As stated in the manuscript, unfortunately the two most amenable to autonomous observation are a very unfavorable combination (i.e. deriving DIC, alkalinity from these comes with relatively large errors). A sensor for alkalinity would significantly improve the situation.

20. Line 929: I think ensemble-based methodologies, in general, provide a useful framework for uncertainty analysis, whether incorporating DA or not.

**Response:** Agree. We see no contradiction to what is written.

Typographical corrections

1. Line 67: An array "is", rather than "are". Alternatively, "Many methods … are …"

**Response:** Changed to "Methods … are …"

2. Line 104: Replace "." by "," after "2.2".

**Response:** For some reason this was fine in the Word doc we uploaded but not in the version that was posted.

3. Line 314: Suggest to replace "alone and" with "alone, which".

**Response:** Sentence has been removed.

4. Line 414: Replace "104" with "$10^4$".

**Response:** Again, this was fine in the Word doc we uploaded but not in the version that was posted.

5. Figure 4: Bottom right text, "desirable".

**Response:** Done. Really appreciate the Reviewer's attention to detail.

6. Line 614: "$pCO_2$" has been used multiple times already and does not need to be defined here.

**Response:** Agree. Definition removed.

7. Line 713: Replace "properties. But" with "properties, but"

**Response:** Done.

8. Line 906: Replace "an" with "and".

**Response:** Done

9. Line 932: Replace "to made" with "to be made".

**Response:** Done

---

## Author Comment (AC3)

**Reviewer 1:**

The manuscript provides a comprehensive review and discussion point about how the biogeochemical coupled hydrodynamic could be used in an Ocean Alkalinization experiments.

The manuscript lays out all the processes and questions any project about OAE should explore. It provides a Guide to Best Practices.

I found the manuscript well written, with only a few points of discussion that could be added, listed below.

**Response:** We appreciate the constructive comments and positive review.

A few dotted points related to the manuscript:

- Any OAE experiment will require the modelling exercise to be complete as soon as the experiment is underway (near real-time) or, even better, in a forecast mode. There are many operational systems available at the global scale but few at the scale of OAE (regional to sub-regional). This is an essential point as the availability of forcing data initial conditions are crucial elements to any modelling system.

**Response:** We agree and added the following statement in section 2.1.2:

*" In this context, model simulations are particularly useful if available in near-real time or in forecast mode. This requires specification of lateral boundary conditions and atmospheric forcing up to the present and into the future. Global 1/12th-degree nowcasts and 10-day forecasts of ocean conditions are available from the Copernicus Marine Service (CMEMS 2023) and atmospheric forcing up to the present and 10 days into the future are available from the European Centre for Medium Range Weather Forecasts (ECMWF 2023)."*

References

ECMWF 2023. European Centre for Medium-Range Weather Forecasts (ECMWF) IFS CY41r2 high-resolution operational forecasts (Accessed on 06-10-2023).

CMEMS 2023. Global Ocean Physics Analysis and Forecast. E.U. Copernicus Marine Service Information (CMEMS). Marine Data Store (MDS). DOI: 10.48670/moi-00016 (Accessed on 06-10-2023)

- In Section 2.2.3. I would add the resuspension due to waves as a significant process in representing the sediment-water exchange. While it only applies to particles, resuspension can be crucial in enhancing carbonate particle dissolution. (Eyre, B. D., Cyronak, T., Drupp, P., De Carlo, E. H., Sachs, J., & Andersson, A. J. (2018). Coral reefs will transition to net dissolving before end of century. *Science*, *359*(6378), 908–911. https://doi.org/10.1126/science.aao1118)

**Response:** We are not aware of evidence that resuspension enhances carbonate dissolution. The reference provided makes no mention of resuspension.

- In section 2.3, I would add a point about the development of unstructured model mesh. Using unstructured mesh allows increasing resolution in a specific area while retaining the ability to have a lower resolution elsewhere to capture larger-scale processes. It can act as an alternative strategy to multiple model nests.

**Response:** We had already mentioned unstructured grid models (line 204) and native grid-refinement (line 551). We have modified the latter sentence as follows to explicitly use the term unstructured grids (new text in bold):

*"Native grid-refinement**, e.g. via unstructured grids,** is another approach that may be pursued to effectively support OAE research."*

- The discussion about the use of DA in OAE could be simplified. While there is room to develop news DA technic to assist OAE, I don't see DA as a major player in OAE.

**Response:** We feel that DA is needed to achieve the most accurate model simulation possible and would like to point to the first comment by the reviewer, where they call for forecast simulations for OAE. We would like to emphasize that forecasts will be more accurate if they involve DA. In section 3.3, we explain (text bolded here for emphasis):

*"Data assimilation (DA) **is the process of improving the dynamical behavior of models** by statistically combining them with observations."*

and

*"Application of DA for ocean models is typically applied for one of two purposes: (1) **to systematically optimize model parameters**, e.g., phytoplankton growth and nutrient uptake or rates of background dispersion, and (2) **to estimate the ocean state**, e.g., distributions of temperature, phytoplankton biomass, alkalinity"*

For separation between OAE signals and natural variability, more accurate, data-assimilative simulations will be much more useful than simulations that display a lot of unconstrained natural variability. Time of emergence, as is commonly used in climate change projections, is likely too imprecise for quantification and verification of OAE impacts. Application of DA will enable more precise attribution. We added the following sentence to tried to improve on this aspect by adding the following text in section 3.3 to articulate this important point:

*"**State estimation offers the potential to constrain variability such that OAE-induced perturbations of carbonate system parameters can be documented even if they are smaller than the natural variability in the study region.**"*

Furthermore, DA methods provide a useful framework for uncertainty quantification, as described in section 3.4.

In the Introduction we modified the text as follows to better explain why DA is helpful and likely needed (new words in bold font):

*"Applications of realistic models rely on them being skillful and accurate, requiring that they include parameterizations of the relevant processes, and that they are constrained by observations that contain sufficient meaningful information **(what is sufficient depends on the application and research question)**. Methods **for constraining models by observations through statistically optimal combination of both** are available. Application of such methods is referred to as data assimilation and provides the most accurate estimates of biogeochemical properties and fluxes."*

- On the other hand, the author could discuss the use of DA in the optimisation phase of any OAE, for example, some of the DA machinery could be used to optimise the amount of Alkalinity being released to maximise the impact area.

**Response:** In our experience, this type of analysis can be done efficiently as a sensitivity analysis. DA feels like overkill in this context.

---

## Author Comment (AC4)

**Reviewer 2:**

I reviewed the original version of this manuscript. I found the paper to be easy to read and given that the paper is a perspective, there was little technical material to evaluate. I have a few suggestions that the authors could consider if they lightly revise the manuscript.

**Response:** We appreciate the constructive comments and positive review.

(1) On line 66, the authors speak to observations that "contain sufficient meaningful information". This seems like a vague and undefined statement. Could you offer some specific examples that help tell the reader what you consider to be "meaningful"?

**Response:** Admittedly information content of observations is an abstract concept. An example of insufficient information content would be comparing the output of a biogeochemical model only to surface chlorophyll observations (or assimilating only surface chlorophyll). This would not be enough to constrain most other state variables and fluxes because the same surface chlorophyll concentration can be achieved by many different combinations of phytoplankton growth and loss rates. A number of examples are given in section 3. We don't want to spell them out with this level of detail in the Introduction because it is supposed to be short and high level, but have modified this sentence as follows (new text in bold):

*"Applications of realistic models rely on them being skillful and accurate, requiring that they include parameterizations of the relevant processes, and that they are constrained by observations that contain sufficient meaningful information **(what is sufficient depends on the application and research question)**."*

(2) Beginning on Line 122: Here, the authors make a seemingly authoritative statement that the direct impacts of OAE on the carbonate system is greatest in the nearfield early in the OAE experiment. Sure, this seems intuitive, but perhaps you could cite literature that supports this statement. Otherwise you could alter the text to reflected that this is an assumption.

**Response:** We are a bit puzzled by this statement because it seems self-evident to us that the biggest impact on the carbonate system is closest to the site of perturbation because mixing and dispersion in the turbulent ocean result in dissipation of the signal away from the perturbation site.

(3) Paragraph on line 418: It might be worth mentioning here that many models in estuaries represent these processes, so I don't think the gap is as big as stated, at least for regional models

**Response:** Two of the authors (Algar and Fennel) work, or have worked, on representing sediment biogeochemical processes in coastal and shelf systems. Both feel strongly that this is indeed a challenge. Again, we are puzzled as to why the statement on line 418, which reads "Representing these processes in coastal and shelf sediments (< 200 m) is challenging." would be controversial.

(4) Line 669-670: It may be worth pointing out here that one might specifically design an observational program that can fully validate the wide-range of impacts of the particular OAE being applied. Perhaps that is implied, but given that different OAE approaches may have different impacts (metals, injections of particulate material), one can tailor their measurements to track the specific impacts of the specific OAE. If $CO_2$ drawdown was all that you cared about, chemical measures of the carbonate system would suffice, but if ecosystem effects were important, you might measure everything the authors describe previously.

**Response:** We agree that an observational program should be designed with the specifics of the OAE application in mind and have expanded the text in section 3.3 as follows:

*"As new platforms are added to the observing system, DA techniques can help guide their optimal deployment **and tailor observational programs to the specific needs of OAE applications** (see Section 4.3 below)."*

We also point to section 3.4 dedicated to OSSEs. We feel that talking about this point in section 3.2 "Validation metrics and approach" would be out of place.